EMBO
reports

# Changing POU dimerization preferences converts Oct6 into a pluripotency inducer

Stepan Jerabek[1] (ID), Calista KL Ng[2], Guangming Wu[1], Marcos J Arauzo-Bravo[3,4], Kee-Pyo Kim[1], Daniel Esch[1], Vikas Malik[5,6,7], Yanpu Chen[5,6,7], Sergiy Velychko[1], Caitlin M MacCarthy[1], Xiaoxiao Yang[5,6,7], Vlad Cojocaru[1,8], Hans R Schöler[1,9,*] & Ralf Jauch[5,6,7,**] (ID)

## Abstract

The transcription factor Oct4 is a core component of molecular cocktails inducing pluripotent stem cells (iPSCs), while other members of the POU family cannot replace Oct4 with comparable efficiency. Rather, group III POU factors such as Oct6 induce neural lineages. Here, we sought to identify molecular features determining the differential DNA-binding and reprogramming activity of Oct4 and Oct6. In enhancers of pluripotency genes, Oct4 co-operates with Sox2 on heterodimeric *SoxOct* elements. By re-analyzing ChIP-Seq data and performing dimerization assays, we found that Oct6 homodimerizes on palindromic *OctOct* more cooperatively and more stably than Oct4. Using structural and biochemical analyses, we identified a single amino acid directing binding to the respective DNA elements. A change in this amino acid decreases the ability of Oct4 to generate iPSCs, while the reverse mutation in Oct6 does not augment its reprogramming activity. Yet, with two additional amino acid exchanges, Oct6 acquires the ability to generate iPSCs and maintain pluripotency. Together, we demonstrate that cell type-specific POU factor function is determined by select residues that affect DNA-dependent dimerization.

**Keywords** DNA binding; Oct4; POU factors; reprogramming to pluripotency
**Subject Category** Stem Cells
**EMBO Reports (2017) 18: 319**–333

## Introduction

In 2006, somatic cells were shown to be reprogrammable to induced pluripotent stem cells (iPSCs) by the overexpression of just four transcription factors (TFs)—Oct4, Sox2, Klf4, and c-Myc (OSKM) [1]. Oct4 is considered to be a unique reprogramming factor, as it could not be replaced by other paralogous members of the POU (Pit-Unc-POU) protein family, while both Sox2 and Klf4 are replaceable and c-Myc can be omitted altogether [2]. Exogenous Oct4 is the most common component of reprogramming mixtures, and activation of endogenous Oct4 is a crucial step in inducing pluripotency. It is unknown which molecular features of Oct4 confer its unique properties, and why other POU factors cannot induce pluripotency in somatic cells.

Oct4 (encoded by the *Pou5f1* gene; reviewed in detail in [3]) is a member of octamer-binding (Oct) TFs, named after the octamer DNA motif with a consensus sequence ATGCAAAT [4–8]. The POU DNA-binding domain has a bipartite structure with two subdomains —the N-terminal POU-specific domain ($POU_S$) and C-terminal POU homeodomain ($POU_{HD}$)—which are connected by a flexible linker region of variable sequence and length among the POU factors [9]. The cooperation between both $POU_S$ and $POU_{HD}$ facilitates proper DNA binding of POU TFs [10], and the linker region further influences the specificity and conformation of the POU–DNA complex [11–13]. The POU factors also possess N- and C-terminal transactivation domains (TADs), which are not conserved among members of this protein family.

Oct4 and other POU factors can bind DNA in versatile modes. Early experimental work done *in vitro* revealed two motifs on which Oct factors can form homodimers. First, two Oct4 molecules need to

---

1   Max Planck Institute for Molecular Biomedicine, Münster, Germany
2   Institute of Medical Biology, Singapore City, Singapore
3   Biodonostia Health Research Institute, San Sebastián, Spain
4   IKERBASQUE, Basque Foundation for Science, Bilbao, Spain
5   Genome Regulation Laboratory, Drug Discovery Pipeline, South China Institute for Stem Cell Biology and Regenerative Medicine, Guangzhou Institutes of Biomedicine and Health, Chinese Academy of Sciences, Guangzhou, China
6   Key Laboratory of Regenerative Biology, South China Institute for Stem Cell Biology and Regenerative Medicine, Guangzhou Institutes of Biomedicine and Health, Chinese Academy of Sciences, Guangzhou, China
7   Guangdong Provincial Key Laboratory of Stem Cell and Regenerative Medicine, South China Institute for Stem Cell Biology and Regenerative Medicine, Guangzhou Institutes of Biomedicine and Health, Chinese Academy of Sciences, Guangzhou, China
8   Center for Multiscale Theory and Computation, University of Münster, Münster, Germany
9   Medical Faculty, University of Münster, Münster, Germany
    *Corresponding author. Tel: +49 251 70365 300; E-mail: office@mpi-muenster.mpg.de
    **Corresponding author. Tel: +86 20 32093805; E-mail: ralf@gibh.ac.cn

bind to a "palindromic octamer recognition element" (*PORE*; ATTTGAAATGCAAAT) for efficient gene activation [14]. Second, POU members can also homodimerize on "more palindromic Oct factor recognition element" (*MORE*; ATGCATATGCAT) [15–17]. The configuration of the bound dimers is substantially different on the *PORE* and *MORE* DNA elements and influences the recruitment of specific cofactors [16].

Further, Oct4 heterodimerizes with alternative partners in the context of different DNA elements. For example, Oct4 dimerizes with Sox2, and the Oct–Sox interface comprises the $POU_S$ of Oct4 and the high-mobility group (HMG) box domain of Sox2 [18–21]. Formation of the Oct4–Sox2 heterodimer is dependent upon the specific DNA element [22]. Genome-wide TF binding studies in ESCs have further authenticated the significance of the Sox2–Oct4 interaction and identified a canonical *SoxOct* element (CATTGTCATGCAAAT) in the enhancers of many pluripotency-related genes, such as *Pou5f1*, *Nanog*, and *Utf1* [23–25]. We had previously reported that Sox17 cooperates poorly with Oct4 on the canonical *SoxOct* element and does not induce pluripotency [26]. However, when a single amino acid at the Oct4 interface of Sox17 was modified, the resultant Sox17EK mutant was found to efficiently cooperate with Oct4 and turns into a powerful iPSC inducer in mouse and human cells [26–29]. A reciprocal Sox2KE mutation eliminates the pluripotency inducing activity of Sox2. Biochemical assays and ChIP-Seq demonstrated that wild-type (WT) Sox17 also cooperates with Oct4, but on an alternative "compressed" DNA element which lacks a single base pair between the *Sox* and *Oct* half sites (CATTGTATGCAAAT). The dimer switch from Sox2–Oct4 to Sox17–Oct4 contributes to the differentiation of ESCs into primitive endoderm [26,28]. The fact that subtle modifications at the molecular interfaces of Sox TFs can profoundly swap their lineage-specifying activities inspired us to ask whether we could identify analogous structural features that are responsible for the function of Oct4.

Here, we compared Oct4 binding motifs to those of other POU factors by re-analyzing ChIP-Seq data and by using quantitative cooperativity assays. We specifically compared Oct4 to Oct6 (encoded by the *Pou3f1* gene), a member of the POU III group. The POU III TFs Brn2 and Brn4 have previously been used for the successful conversion of mouse and human cells into neurons and neural precursor cells [30–37]. Interestingly, for cells undergoing lineage conversions triggered by OSKM and BSKM (Brn4 instead of Oct4), a transient Oct4-positive state was recently described [38]. So far, Oct6 was not used for any lineage conversion. Moreover, Oct6 does not appear to be capable of generating iPSCs [2]. Here, we report differences between Oct4 and other POU TFs in dimerizing on composite DNA motifs as a means to direct specific cell fate choices and inducing pluripotency.

# Results

## Oct transcription factors differentially bind enhancer signature motifs

To investigate the basis for reprogramming activity of specific POU family proteins, we re-analyzed publically available ChIP-Seq data sets in order to discover enriched DNA motifs *de novo* in various cell types [39]. As expected, mouse ESCs showed a marked enrichment

of the *SoxOct* composite motif (*P*-value 1e-7,149). In contrast, all analyzed somatic cells revealed the palindromic *MORE* among the top scoring motifs including Oct2 in B cells (1e-218), mouse embryonic fibroblasts (MEFs) after 48 h of transfection with Brn2 (1e-1,120) and Brn2 in unipotent/oligopotent mouse neural progenitor cells (mNPCs, *P*-value 1e-1,234) (Fig 1A). To further determine the abundance of the *SoxOct* heterodimer and *MORE* homodimer motifs, we performed position weight matrix (pwm) scanning as well as text search using IUPAC strings corresponding to *SoxOct* and *MORE* sequences in the ChIP-Seq-enriched regions. Consistent with *de novo* motif discovery, these analyses show that the *MORE* motif predominates in somatic Oct binding sites, whereas the *SoxOct* motif is strongly enriched in the Oct4 binding sites of pluripotent cells (Fig EV1A). Structural models illustrate the profound topological differences in DNA-bound Sox–Oct heterodimers and Oct–Oct homodimers (Fig 1B and C), and we speculated that these differences contribute to the formation of disparate enhanceosomes in pluripotent and somatic cells.

To study the differential binding preferences of POU TFs *in vitro*, we performed quantitative electrophoretic mobility shift assays (EMSAs) using purified POU domains and compared binding of the Oct4 POU and the Oct6 POU to *MORE* DNA (Fig 1D). Homodimeric Oct4/*MORE* complexes are prominent only at high protein concentrations when the free DNA becomes limiting. In contrast, Oct6 POU/*MORE* dimer bands are clearly visible from the beginning of the titration series, demonstrating higher cooperativity of Oct6 compared with Oct4. Next, we extended the analysis and included a total of six TFs belonging to POU classes I, II, III, V, and VI (Figs 1E and F, and EV1B). We quantified the cooperativity of the homodimeric Oct–Oct and heterodimeric Sox–Oct binding of the POU proteins to the *MORE* (*OctOct*) and *SoxOct* elements using previously derived equations [27,40]. Cooperativity represents the ratio of the equilibrium binding constant for the binding of a factor to free DNA to the binding of DNA in the presence of another protein (Sox2 for heterodimers and another POU factor for homodimers). The higher the cooperativity, the more one protein facilitates the binding of a second protein. In accordance with our *in silico* analysis, we found that the Oct4 POU only weakly dimerizes on the *MORE*, whereas all other POU factors examined strongly cooperate on this element (Fig 1E). The differences in the *SoxOct* element are less pronounced, but we observed ~twofold preference for the formation of the Sox2–Oct4 complex. Nevertheless, Pit1, Oct1, Oct6, and Brn2 can also form heterodimers on the *SoxOct* element (Fig 1F). Brn5 is the only POU factor that could not heterodimerize with Sox2.

We further verified the rebalanced binding pattern of Oct4 and Oct6 in single-tube EMSAs. In these experiments, POU factors and Sox2 are simultaneously incubated with FAM-labeled *MORE* and Cy5-labeled *SoxOct* DNA elements and the equilibrium abundance of FAM and Cy5 DNA is quantified using successive fluorescent scans (Fig EV1C and D). Consistent with the cooperativity measurements, plotting the ratios of heterodimers (Cy5 scan) and homodimers (FAM scan) bands revealed a relative preference of Oct4 for the *SoxOct* motif and a preference of the other POUs for the *MORE* motif (Fig EV1D). The differential propensity of Oct4 and somatic POU factors to associate with *SoxOct* and *MORE* DNA elements could contribute to their functional uniqueness. One interesting possibility is that the two dimeric states are part of a *cis*-regulatory system helping to establish and/or maintain different cell fates.

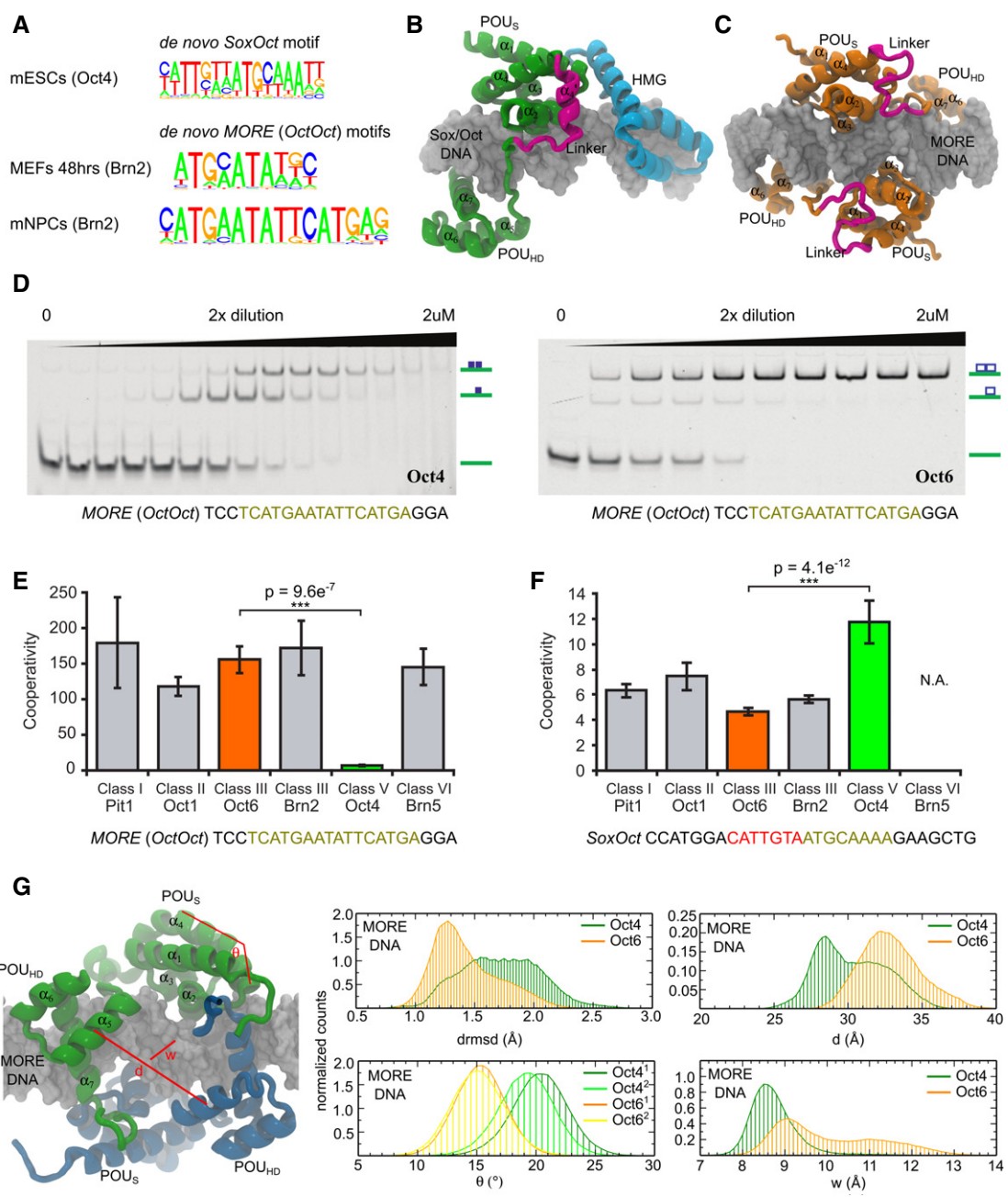

**Figure 1.  Oct transcription factors exhibit differential preferences for composite DNA-binding sites.**

A    Position weight matrices obtained using *de novo* motif discovery (HOMER) and ChIP-Seq summits of Oct4 in mouse ESCs [56], Brn2 in MEFs 48 h after transfections, and Brn2 in mouse NPCs [23].

B, C    Molecular models based on the crystal structures previously published [44] represent configurations of two dimers on DNA: Sox2–Oct4 heterodimer (B) and Oct6–Oct6 homodimer (C). Oct4 POU domain, green; Oct4/Oct6 linker region, magenta; Sox2 HMG, cyan; Oct6 POU domain, orange. Individual helixes of POU domains are numbered. In both models, the DNA is represented as a light gray surface. Unless indicated otherwise, the representation is kept throughout the manuscript.

D    EMSAs with a twofold dilutions series starting at 2 µM of the Oct4 POU (left) or the Oct6 POU (right) in the presence of Cy5-labeled *MORE* DNA. The free DNA and DNA bound as monomer or dimer are indicated.

E, F    Bar plots of EMSA-derived cooperativity factors for Oct–Oct homodimers on the *MORE* (E) or Sox2–Oct heterodimers on the *SoxOct* element (F) are shown for a panel of six POU proteins. The mean is shown with standard deviation as error bars ($n \geq 3$), and Tukey's multiple comparison of means was used for assessment of statistical significances (***$P < 0.001$). N.A.: not assessed.

G    Properties of Oct4 and Oct6 homodimers from molecular dynamics (MD) simulations. The structural snapshot on the left illustrates Oct4 (first monomer in green, second in dark blue) and the three parameters used: drmsd = the root-mean-square deviation of interatomic distances measured for the backbone of the structured domains (excluding the linker); $\theta$ = the angle describing the kink in helix $\alpha_4$ (data for both monomers 1 and 2 are shown); d = the distance between the centers of mass of the two helices $\alpha_5$; w = the DNA minor groove width in the region between the two monomers.

Source data are available online for this figure.

## The Oct4 homodimer is unstable and structurally flexible

Besides equilibrium binding, the kinetics of TF–DNA interactions is important to elicit regulatory responses [41]. In particular, the residence time on *cis*-regulatory DNA sequence is critical. We therefore tested whether the Oct6 and Oct4 POU domains assemble with *MORE* and *SoxOct* DNA with a different dissociation kinetics using competition EMSAs. Strikingly, the Oct6 homodimer on *MORE* DNA is substantially more stable than both Oct6–Sox2 and Oct4–Sox2 heterodimers on *SoxOct* elements (Fig EV2A–D). The Oct4 homodimer dissociates more quickly than the Oct6 homodimer from *MORE* DNA. Therefore, the formation of short-lived Oct4/*MORE* versus long-lived Oct6/*MORE* homodimers likely contributes to the unique function of the two POU factors. To investigate the structural basis of the observed differences, classical molecular dynamics simulations were performed demonstrating that both Oct4 and Oct6 homodimers stably bound the *MORE* DNA during several hundred ns-simulated timescale. However, Oct4 showed more flexibility as measured by the root-mean-square deviation of the interatomic distances (drmsd) of the backbone of the structured regions (Fig 1G). Moreover, in Oct4, the kink in helix $\alpha_4$ (POU$_S$) induced by Pro70 (angle $\theta$) was more pronounced and a concerted motion of the two POU$_{HD}$ domains (distance d) induces a narrowing of the minor groove (width w) in the region of the DNA between the two monomers (Fig 1G). These findings provide a structural-based explanation for the less favorable homodimer geometry in Oct4.

## Rebalancing Oct homo- and heterodimerization with a single amino acid swap

We next sought to identify structural determinants of the differential homodimerization on *MORE* DNA using the crystal structures of Oct1 and Oct6 [17,42] and structural model of Oct4 on *MORE* DNA. Residue 151 mediates intermolecular contact between the POU$_{HD}$ and the POU$_S$ [42]. This residue maps to the C-terminal part of the DNA recognition helix 7 of the POU$_{HD}$ and encodes a Met in the POU III group or other aliphatic residues (Val or Ile) in the POU I and POU II groups, respectively. In contrast, Oct4 encodes a polar Ser at position 151 (Figs 2A and EV3A). We inspected the chemical environment of residue 151 in structural models and during molecular dynamics (MD) simulations. In Oct6 and Oct4$^{S151M}$, Met151 docks in a hydrophobic pocket of the POU$_S$ domain of the second monomer (Fig 2A). MD simulations revealed a network of stable hydrophobic interactions around Met151 in Oct6 as shown by the minimal interatomic distances between the residues involved (Fig 2B). In Oct4, the hydrophobic pocket is closed by a concerted motion leading to the more pronounced kink in helix $\alpha_4$ (Fig 1G). Specifically, Leu9 ($\alpha_1$ of the POU$_S$) and Phe62 ($\alpha_4$ of the POU$_S$) move closer to the center of the pocket to compensate for the smaller, hydrophilic side chain of Ser151 establishing a less optimal network of interactions (Fig 2C). We therefore reasoned that residue 151 constitutes a key determinant guiding homodimeric DNA recognition by POU TFs.

To test this hypothesis, we exchanged this amino acid to construct mutated Oct4$^{151M}$ and Oct6$^{151S}$ proteins (Fig EV3B). To examine dimerization of mutant POU factors on the *MORE,* we performed EMSAs with the different variants (Fig 2D and E). Strikingly, we found that Oct4$^{151M}$ gains the ability to homodimerize on

the *MORE* with a cooperativity similar to that of the WT Oct6 POU domain (Fig EV3C). In contrast, the reciprocal Oct6$^{151S}$ mutation reduced the cooperativity toward the level of WT Oct4. However, binding to the *SoxOct* element is not substantially altered, as both mutant proteins retain the ability to heterodimerize with Sox2 (Figs 2E and EV3C). Consistently, the ratio of heterodimers and homodimers in single-tube EMSAs shifted toward the *MORE* for Oct4$^{151M}$ and toward *SoxOct* for Oct6$^{151S}$ (Figs 2F and EV3D).

In summary, we identified a single amino acid residue in helix 7 of the POU$_{HD}$ of Oct factors determining the discrimination between *SoxOct* and *MORE* DNA (Fig 2G).

## Identification of Oct4 elements critical for pluripotency induction

We next asked whether modifications in *MORE* and *SoxOct* DNA recognition have an impact on global transcriptional programs and, ultimately, on cellular fate decisions. To address this, we systematically probed structural elements that might be responsible for the uniqueness of Oct4 during somatic cell reprogramming. For this analysis, four structural elements were selected as illustrated by the structural models in Fig 3A and B. First, the amino acid influencing homodimerization on the *MORE* was mutated to generate Oct4$^{151M}$. Second, we chose a double mutant in the first alpha helix of the Oct4 POU$_S$ subdomain (Oct4$^{7D,22K}$) previously shown to be required for maintaining pluripotency [43]. Third, a double mutation that completely abolishes Oct4 interaction with Sox2 on canonical *SoxOct* elements was generated [17] (Oct4$^{21Y,29R}$). Fourth, the POU domain linker of Oct4 was replaced by that of Oct6 (Oct4$^{LinkO6}$). Altogether, we generated 10 Oct4-based constructs (Fig 3C). Except for the SoxOct mutation designed to abolish the interaction between Sox2 and Oct4, we modified the selected residues by replacing them with their Oct6 counterparts. Of note, our structure-based linker alignment differs from the one published previously [44]. The difference is caused by a low conservation of the N-terminal part of the linker, and as a result of the new alignment, Oct6 has one additional positive charge in its "RK" region following the POU linker (Fig EV3A, Appendix Fig S1A–C).

After assessment of viral titers by qRT–PCR (Appendix Fig S2A), we infected mouse embryonic fibroblasts (OG2-MEFs) with the combination of retroviruses carrying an Oct4 variant together with three other mouse factors—Sox2, Klf4, and c-Myc. The assay was evaluated 16 days after the infection. The plates for all combinations of transcription factors were screened for the presence of Oct4-GFP-positive colonies under fluorescence microscope (Appendix Fig S2B), and the efficiency was plotted as the number of GFP-positive colonies for each condition (Fig 3C). Oct4$^{151M}$ and Oct4$^{7D,22K}$ generated GFP-positive colonies (Fig 3D, Appendix Fig S2B) with a colony yield of about 60% compared to WT Oct4. When the Sox2–Oct4 interface was completely disrupted (Oct4$^{21Y,29R}$), we observed no GFP-positive colonies at all (Fig 3C and D, Appendix Fig S2B). This confirms the importance of Sox2–Oct4 heterodimerization, in agreement with recent work revealing the relevance of distinct Sox2–Oct4 heterodimer configurations for induction and maintenance of pluripotency [45]. Interestingly, the Oct4$^{SoxOct/151M}$ mutant induced a small number of GFP-positive colonies, suggesting that the 151M mutation rescued some of the detrimental effects introduced by the Oct4$^{21Y,29R}$. When the Oct4 linker region was replaced

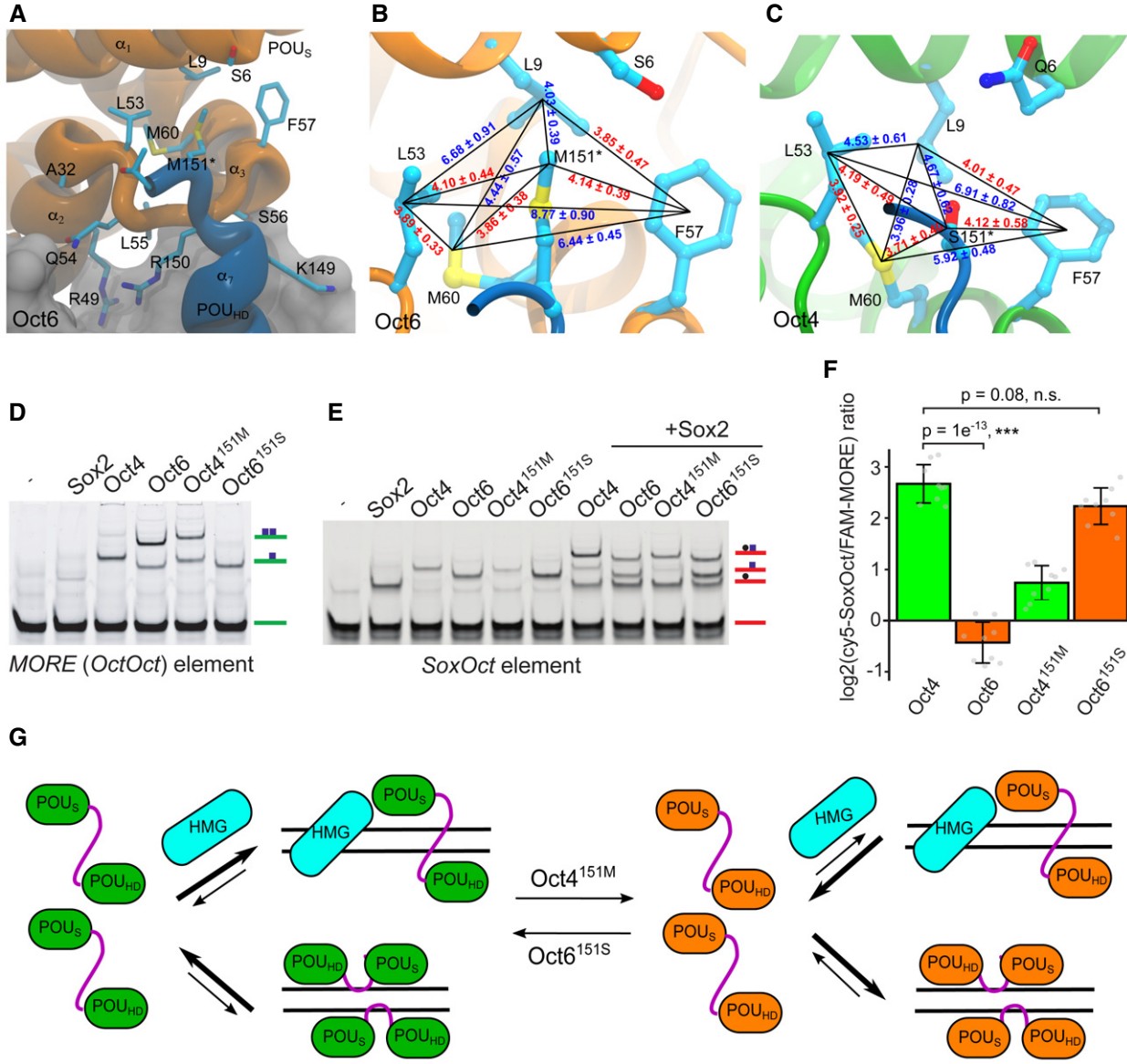

**Figure 2. A single amino acid residue guides DNA-binding preferences of Oct4 and Oct6 POU domains.**

A   Zoom-in view of the Oct6–Oct6 homodimer interface in the molecular models based on crystal structures of two POU homodimers on *MORE* DNA. Oct6 POU$_S$, orange; POU$_{HD}$ of the other monomer, blue. Important residues are labeled and Met (Oct6) in position 151 that was identified to guide the DNA-binding preference is marked with a star. Individual helices of POU domains are numbered. Color representation as in Fig 1B and C. For clarity, the second monomer of Oct4/Oct6 is shown in dark blue.

B, C   Network of hydrophobic interactions in Oct6 (B) and Oct4 (C). The numbers show the minimal interatomic distances (excluding hydrogens) between the residues involved calculated from the simulations. In blue are those distances that change significantly between Oct4 and Oct6 and in red those unchanged. The shown standard deviations represent a measure for the variability during the simulations and not for the uncertainty in the calculations.

D   EMSA showing a differential binding of WT Oct4 and Oct6 POU domains as well as mutated Oct4$^{151M}$ and Oct6$^{151S}$ POU domains on *MORE* (*OctOct*) element. The free DNA and DNA bound by monomer or dimer are indicated.

E   EMSA showing binding of WT Oct4 and Oct6 POU domains as well as mutated Oct4$^{151M}$ and Oct6$^{151S}$ POU domains to *SoxOct* element in the absence or presence of the Sox2 HMG. The free DNA and DNA bound by monomers or dimer are indicated. Quantifications of cooperativity measurements are shown in Fig EV3C.

F   Bar plot based on single-tube EMSAs using Cy5-*SoxOct* and FAM-*MORE* DNA elements showing the difference in DNA binding of WT Oct4, WT Oct6, Oct4$^{151M}$, and Oct6$^{151S}$ POU domains. The mean log2 ratios of Cy5 (heterodimer) and FAM (homodimer) band intensities are depicted with standard deviation ($n = 9$). Individual data points are shown as gray jitter plots. Tukey's multiple comparison of means was performed to assess significance (***$P < 0.001$, n.s., $P > 0.05$). Compare to Figs EV1D and EV3D.

G   A scheme illustrating the DNA-binding preferences of Oct POU domains to *MORE* and *SoxOct* elements. On the left side, WT Oct4 (green) preferentially heterodimerizes with Sox2 HMG (cyan) over homodimerizing on MORE. In the second scenario, WT Oct6 (orange) preferentially forms homodimers on *MORE* DNA rather than heterodimers with Sox2 on *SoxOct* DNA elements. The thickness of the arrows illustrates the relative abundance of the microstates under equilibrium conditions. The binding behavior is swapped for the engineered Oct6$^{151S}$ and Oct4$^{151M}$ proteins.

Source data are available online for this figure.

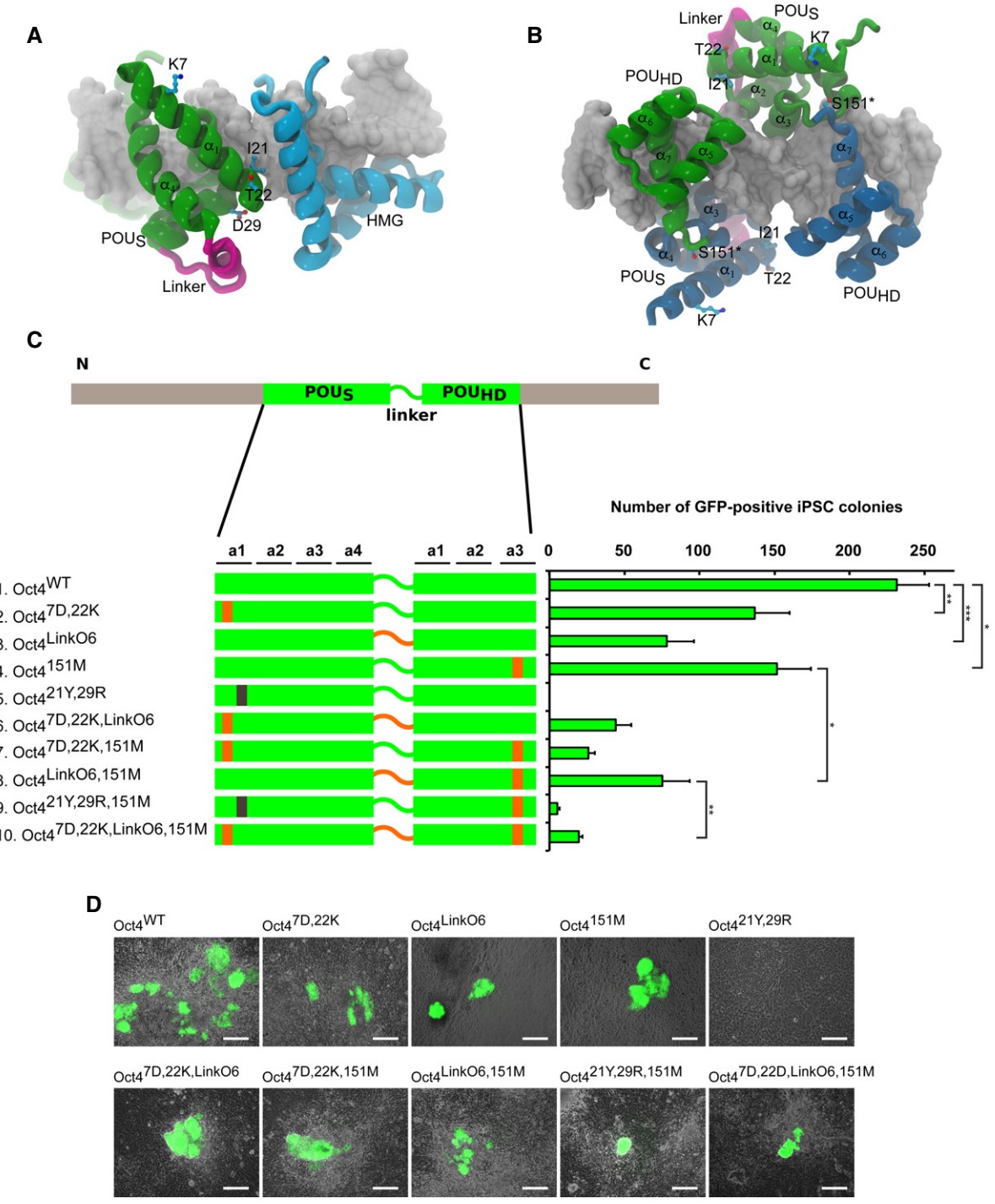

**Figure 3.  The residue guiding DNA-binding preference is a key Oct4 determinant in pluripotent cell generation.**

A, B   Molecular models show the position of mutated residues in the structures of two dimers on DNA: Sox2–Oct4 heterodimer (A) and Oct4–Oct4 homodimer (B). Ser in position 151 (asterisk) was mutated to Met in order to shift the preference of Oct4 for homodimerization. The individual helixes of the POU domains are numbered.

C   On the left side, a schematic overview of Oct4 and its mutants used for iPSC generation. $POU_S$ and $POU_{HD}$ are shown as green bars connected by the linker. Sites mutated to the respective Oct6 residues and Oct6 linker are denoted in orange, and the mutation in the Sox–Oct interface is in black. The efficiency of these constructs for iPSC generation from MEFs is depicted as the absolute number of GFP-positive colonies on the right. Error bars represent standard deviations of biological replicates ($n = 3$), and differences between selected samples were compared using ANOVA (P-values: $Oct4^{WT}xOct4^{7D,22K}$ $P = 6.9e-3$; $Oct4^{WT}xOct4^{LinkO6}$ $P = 7.3e-4$; $Oct4^{WT}xOct4^{151M}$ $P = 1.2e-2$; $Oct4^{151M}xOct4^{LinkO6,151M}$ $P = 1.1e-2$; $Oct4^{LinkO6,151M}$ $xOct4^{7D,22K,LinkO6,151M}$ $P = 7.1e-3$) (***$P < 0.001$, **$P < 0.01$, *$P < 0.05$).

D   GFP-positive colonies of mouse iPSCs generated by Oct4 mutants in combination with Sox2, Klf4, and c-Myc. Colonies were imaged 16 days after second viral infection, using a fluorescence microscope. Scale bars: 250 μm; 10× objective.

Source data are available online for this figure.

by its Oct6 counterpart (Oct4$^{LinkO6}$), the iPSC generation efficiency dropped to 30% of the WT level (Fig 3C). Additionally, we performed reprogramming experiments using two different Oct4$^{LinkO6}$ constructs side by side; one construct was based on a sequence alignment of the linkers and one on a structure-based alignment (Appendix Fig S1C). This experiment showed that when the entire Oct4 and Oct6 RK motifs are aligned with the central gap as the structural alignment suggests, the Oct4$^{LinkO6}$ devoid of one positively charged "R" is able to induce pluripotency, albeit with reduced efficiency (Appendix Fig S1C).

Overall, our reprogramming experiments provided evidence for important roles of rationally selected Oct4 modifications for the generation of iPSCs.

## An engineered Oct6 is capable of induction and maintenance of pluripotency

As WT Oct6 cannot induce pluripotency, we asked whether elements critical for Oct4 function would enable the conversion of Oct6 into a reprogramming factor (Fig 4A). We first adjusted the concentration of viruses (Fig EV4A) and confirmed that Oct6 cannot replace Oct4 in somatic cell reprogramming, as an O6SKM cocktail did not produce GFP-positive colonies. Moreover, swapping of the four structural elements one by one was insufficient to convert Oct6 into a reprogramming TF. However, when 151S was combined with the 7K, 22T double mutant identified by Nishimoto et al to be critical for pluripotency maintenance [43], we consistently obtained iPSC colonies (Figs 4A and EV4B). Colony yield further increased when the Oct6 linker region was replaced with its Oct4 counterpart. We therefore established two stable iPSC lines generated with this engineered Oct6 (Oct6$^{7K,22T,LinkO4,151S}$; O6SKM iPSCs1 and O6SKM iPSCs2) for further characterization (Fig 4B).

Next, we confirmed the absence of Oct4 transgene (Fig 4C) and verified the efficient silencing of all four viral transgenes in O6SKM cells (Fig 4D). Moreover, both O6SKM lines were karyotypically normal (Fig EV4C). We then asked whether the O6SKM iPSCs possess all hallmarks of pluripotency. First, we confirmed the expression of both endogenous Sox2 and Nanog in O6SKM iPSCs by immunochemistry (Fig 4E). Next, we determined the DNA methylation status of *Oct4* and *Nanog* promoter regions by bisulfate sequencing using the collagen type I alpha 1 (*Col1a1*) locus as a control. CpG methylation present in MEFs was completely removed from the *Oct4* and *Nanog* regulatory regions in both O6SKM iPSC lines like in mouse ESCs (Fig 4F). Finally, we compared the global gene expression profiles of O6SKM iPSCs to O4SK iPSCs, bona fide OG2-ESCs, as well as previously published O4SKM cells and MEFs (Figs 4G and EV4D). Overall, all pairwise scatter plots as well as the hierarchical clustering demonstrated that the global gene expression profile is highly similar to standard O4SK iPSCs and ESCs, but substantially different from the profile of MEFs (Figs 4G and H, and EV4D).

In order to assess the differentiation potential of O6SKM iPSCs, we performed two assays. In one assay, we generated embryoid bodies (EBs) via the hanging-drop method. After immunostaining of the spontaneously differentiated EBs, we verified the expression of markers typical for all three germ layers (Fig 5A). In the other assay, O6SKM iPSCs were injected subcutaneously into severe combined immunodeficiency (SCID) mice to see whether they had

the potential to form teratomas. Indeed, cells from both O6SKM iPSC lines gave rise to teratomas and cells of all three germ layers could be detected (Fig EV5A). Finally, the developmental potential was evaluated by the aggregation assay. For this, we aggregated the O6SKM iPSCs with diploid mouse embryos. Once the aggregates developed into blastocysts, they were transferred into recipient mice. The ability of O6SKM iPSCs to contribute to the germline was demonstrated after observation of Oct4-GFP signal in the gonads of E13.5 fetuses and E19.5 pups (Fig EV5B and C). We genotyped the tails of chimeras (Fig EV5D) and confirmed the presence of the Oct6 transgene (Fig EV5E). To test for germline transmission, we mated the chimeric mice with foster mothers. In the newborn pups (Fig 5B), we observed GFP signal in the ovaries and testes (Fig 5C). We subsequently also tested for the presence of Oct6 transgenes by PCR and found that the viral transgene could be readily amplified (Fig EV5F).

Last, we asked whether Oct6$^{7K,22T,LinkO4,151S}$ can maintain pluripotency using a complementation assay and the transgenic ZHBTc4 ES cell line [43]. In this assay, endogenous Oct4 is suppressed upon doxycycline addition and pluripotency can only be rescued if additional exogenous factors are provided. Exogenous WT Oct4 is able to rescue pluripotency, while WT Oct6 cannot [43]. It was found that even a low level of Oct6$^{7K,22T,LinkO4,151S}$ can rescue pluripotency and sustain the expression of pluripotency genes (Fig EV5G and H). ESCs rescued with Oct6$^{7K,22T,LinkO4,151S}$ exhibit the typical colony morphology and self-renew, albeit with reduced proliferation rate when compared to WT Oct4-rescued cells.

In summary, our engineered Oct6 produces iPSCs passing all *in vitro* and *in vivo* assays validating their pluripotency and is also able to maintain the pluripotency of ESCs.

# Discussion

### Oct4 has unique DNA-binding preferences

Oct4 is a master regulator of pluripotency, and it is the only factor that cannot be substituted by any paralogous family member during iPSC generation [2]. What then are the unique molecular features of this POU factor endowing it with this capacity? Previous studies have provided evidence that the DNA-binding POU domain should be at the center of our search. For example, constructs lacking either C- or N-terminal TAD still maintained a stem cell phenotype and yielded high numbers of ESC colonies in the Oct4 complementation assay [46]. Moreover, an unbiased Ala scan along the Oct4 molecule and subsequent reprogramming indicated that the POU domain—but not the N- and C-terminal TADs—is the most critical part of the protein for iPSC generation [47]. Furthermore, the N- and C-termini of Oct4 could be replaced with the TAD of the Yes-associated protein (YAP) without loss of reprogramming activity [47]. However, introduction of only two mutations into helix 1 of the POU$_S$ led to a dramatic decrease in ability of Oct4 to maintain pluripotency *in vitro* [43].

After re-analyzing a panel of ChIP-Seq data for the POU domain protein, we noticed that Oct4 binds the *SoxOct* motif, whereas somatic POUs preferentially form homodimers on palindromic binding sites (Figs 1A and EV1A). Neural POU proteins have previously been reported to form highly cooperative homodimers *in vitro* [48].

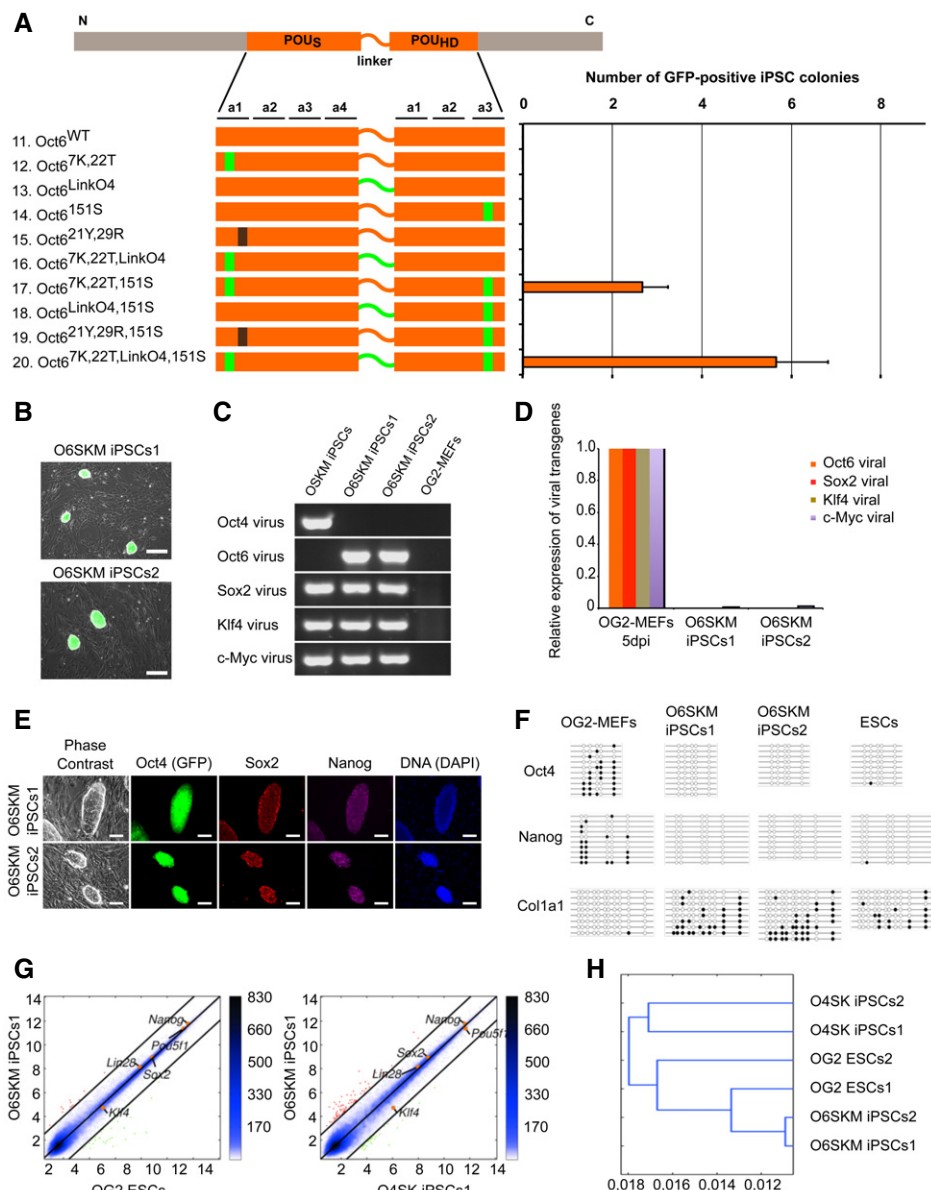

**Figure 4. The synthetic Oct6 molecule contributes to epigenetic reprogramming of mouse embryonic fibroblasts.**

A   On the left side, a schematic overview of Oct6 and its mutants used for iPSC generation. POU$_S$ and POU$_{HD}$ are shown as orange bars connected by the linker. Sites mutated to the respective Oct4 residues and Oct4 linker are denoted in green, and the mutation in the Sox–Oct interface is in black. The efficiency of these constructs for iPSC generation from MEFs is depicted as the absolute number of GFP-positive colonies on the right. Error bars represent standard deviations of two biological replicates run in parallel.

B   Oct4-GFP-positive colonies of two O6SKM iPSC lines, expanded from single colonies of pluripotent cells that were generated by the contribution of the synthetic Oct6 molecule (Oct6$^{7K,22T,LinkO4,151S}$). Scale bars: 250 μm.

C   Genotyping of two O6SKM iPSC lines. Two stable lines of iPSCs generated with the Oct6, Sox2, Klf4, and c-Myc retroviral set are positive for these four transgenes, but negative for the Oct4 transgene. OSKM iPSCs and OG2-MEFs were used as PCR controls.

D   Expression analysis of viral transgenes in two O6SKM iPSC lines done by qRT–PCR using specific primers. OG2-MEFs were harvested 5 days after infection and used for comparison as a positive control.

E   Two O6SKM iPSC lines immunostained for pluripotency markers Sox2 and Nanog. Oct4 expression was confirmed by the Oct4-GFP transgene. DNA stained by DAPI. Scale bars: 100 μm.

F   Bisulfite sequencing of genomic *Oct4*, *Nanog*, and *Col1a1* promoter regions of O6SKM iPSCs, OSKM iPSCs, and OG2-MEFs. White and black circles represent unmethylated and methylated CpG sites, respectively.

G   Pairwise scatter plots comparing the global gene expression profile of O6SKM iPSCs1 with OG2 ESCs (left) and O4SK iPSCs (right). Black lines represent a twofold change in gene expression levels between the paired cell lines. On the right side of the plots, the color bar indicates scattering density. Red and green dots represent up- and downregulated genes, respectively. Positions of selected pluripotency-related genes are highlighted as orange points.

H   Hierarchical clustering of O6SKM iPSC lines with OG2 ESCs and O4SK iPSCs, illustrating the close relationship between their global gene expression profiles.

Source data are available online for this figure.

   

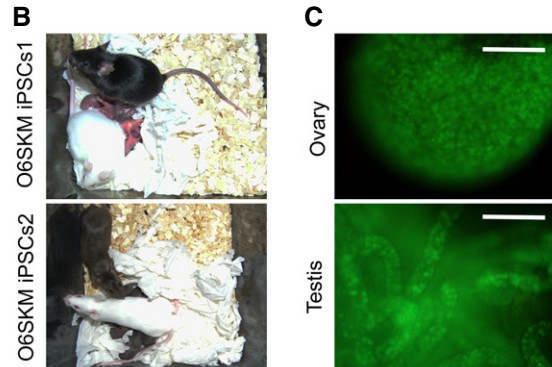

**Figure 5. Cells reprogrammed using the synthetic Oct6 molecule show pluripotency *in vitro* and *in vivo*.**

A *In vitro* differentiation of O6SKM iPSC lines into cells of all three germ layers as shown by immunochemistry: endoderm (α-fetoprotein, AFP), mesoderm (α-smooth muscle actin, SMA), and ectoderm (β-tubulin, TUJ1). Nuclei (DNA) were stained by Hoechst (blue). Scale bars: 200 μm.

B F1 offspring of chimera male mice with contribution from O6SKM iPSCs and CD1 female mice.

C Pictures demonstrating germline transmission; endogenous Oct4-GFP signal was detected in the gonads of F1 pups. Scale bars: 200 μm.

A recent study has shown that class III POU TFs preferentially target the *MORE* sequence in NPCs [49]. The authors used full-length proteins fused to large fluorescent tags to verify that Oct6 preferentially forms homodimers, whereas Sox2 preferentially heterodimerizes with Oct4 on *SoxOct* elements. However, diminished binding of Oct4 to the *MORE* DNA element *per se* was not reported. We extended this work by using untagged DNA-binding domains and quantitative cooperativity measurements of a panel of five POU proteins. Our results show that Pit1, Oct1, Oct6, Brn2, and Oct4 all show positive cooperativity on the canonical *SoxOct* motif (Fig 1F). In contrast, the binding mode on the *MORE* DNA element is markedly different, as the cooperativity and the residence time of Oct4 are profoundly reduced (Figs 1E and EV2). We therefore surmise that the reduced cooperativity on the *MORE* DNA element contributes to the unique mechanism setting Oct4 apart from other POU factors (Fig EV1C and D).

### Modulating Oct4 and Oct6 DNA recognition and its relevance for the interaction with cofactors

In our search for the structural basis of binding differences in the *MORE* DNA element, we identified candidate elements in the crystal structures of Oct1 and Oct6 TFs bound to this DNA motif [17,42]. By exchanging a single residue between Oct4 and Oct6, we swapped their DNA-binding preferences. Our *in vitro* experiments showed that the Oct6[151S] mutant is re-distributed from *MORE* to *SoxOct* elements, while binding of the mutated Oct4[151M] shifts toward homodimers on the *MORE* DNA element (Fig 2D and E). Moreover, after changing the binding properties of the Oct4 mutant, we observed a drop in its capability for iPSC generation to about 65% of the WT protein. Conversely, Oct6 could only generate iPSCs if the ability to homodimerize on the *MORE* was decreased with a reciprocal mutation (Figs 4A and EV4B). Therefore, rebalancing the formation of heterodimeric and homodimeric complexes substantially alters the ability of POU factors to direct cell fate decisions.

A previous study has shown that the class I POU factor Pit1 can either activate the growth hormone gene or cause its effective suppression in a cell type-dependent manner [50]. This switch of a

biological outcome is triggered by a two-base pair difference in the promoter DNA. Apparently, allosteric changes to the conformation of the POU domain induced by the DNA sequence influence the factor's interaction with a specific repressor complex [50]. Similarly, another seminal study demonstrated that the B-cell-specific coactivator OBF1 interacts with Oct factor dimers on *PORE* DNA. In contrast, Oct dimers bound to the *MORE* DNA element do not recruit OBF1 [16]. We thus surmise that the differential association of Oct6 and Oct4 with *SoxOct* versus *MORE* DNA element not only affects which genomic loci are targeted, but also which set of cofactors is recruited and how the expression of nearby genes is regulated. Testing this hypothesis requires the examination of the interactome of Oct4 and Oct6 in the context of specific enhancer signatures.

### Multiple functions of the POU linker region

One of surprising observations in our study was that a chimeric protein in which the Oct4 linker is substituted by the Oct6 linker (Oct4[LinkO6]) still retains 30% of the iPSCs inducing efficiency (Fig 3C and D). This result contrasts former observations reporting no iPSC colonies for Oct4[LinkO6] constructs [43,44]. This discrepancy can be explained by a structural alignment of the linker sequences, in which the entire RK motifs of Oct4 and Oct6 are aligned with a central gap causing an Arg residue to be part of the swapped linker in one but not the other construct (Appendix Fig S1A). Recently, an interplay between the linker segment and the RK motif following the linker has been described and shown to affect not only DNA binding but also transactivation potential and reprogramming efficiency [13]. However, whether the two different POU linkers interact with distinctive epigenetic modifiers remains unknown.

### Cooperativity between exogenous Oct4 and Sox2 during the initiation of reprogramming?

Mutations in the Sox–Oct interface of Oct4 (Oct4[7D,22K]) led to considerable drop in reprogramming efficiency, demonstrating that uncoupling of Oct4 from Sox2 (Oct4[21Y,29R]) has a profound

detrimental effect on iPSC generation (Fig 3C and D). A study using single-molecule imaging recently showed a sequential engagement of TFs to their target sites in ESCs. The authors showed that Sox2 binds to DNA first and subsequently recruits Oct4 to assemble the Sox–Oct heterodimer [51]. Possibly, this recruitment process is impaired for the Oct4[21Y,29R] protein. Moreover, Oct4 seems to possess an intriguing ability to bind a closed chromatin as a pioneer factor and to initiate chromatin opening [52]. Our results strongly suggest that cooperation between Oct4 and Sox2—and not Oct4 alone—is involved in not only maintaining but also establishing pluripotency. Whether and how this cooperation affects opening of the chromatin remain elusive.

### Outlook

Synthetic biology is emerging as a provider of powerful tools for cellular reprogramming and stem cell biology with a focus on the CRISPR-Cas-based genome engineering. In addition, new opportunities arise to design artificial TFs with enhanced biological properties (reviewed in [53]). Our identification of critical elements underlying POU function and determining the unique properties of Oct4 can guide the generation of synthetic TFs that direct cellular identities as an alternative to CRISPR-Cas-based TF design.

# Materials and Methods

### ChIP-Seq analysis

In order to compare the occurrence of *OctOct (MORE)* versus *SoxOct* motifs bound by Oct2, Brn2, or Oct4 in different cell types, we downloaded ChIP-Seq data from the GEO database. Oct2 peaks in B cells (GSE21512) [39], Brn2 in MEFs after 48 h of reprogramming (GSE43916) [54], Brn2 in NPCs (GSE43916) [54], (GSE35496) [55], and Oct4 in ESCs (GSE11724) [56]. ChIP-Seq reads were aligned to the mouse genome (mm10 assembly) using bowtie2 [57]. TF binding peaks were called using MACS [58] with the respective input data as control. As the SRA files from Oct2 ChIP-Seq in B-cell study were not available, we used peak coordinates provided by the authors and used UCSC batch coordinate conversion (liftOver) function to convert mm8 coordinates to mm10. Motif analysis of Oct2, Brn2, and Oct4 binding at either the *MORE* or *SoxOct* motif was done using HOMER findMotifsGenome.pl [39]. Motif occurrences were calculated within 200-bp windows centered on ChIP-Seq summits by using FindMotifsGenome.pl with options –find and –len 12,14,16 using PWMs discovered from *de novo* motif analysis. Motif counting was also performed using word search in R (https://www.r-project.org/) where genomic locations were converted to genomicRanges objects and sequences were retrieved from mm10 genome version using the BSgenome.Mmusculus.UCSC.mm10 object and the getSeq function (Biostrings). *SoxOct* and *MORE* motif searches were performed using IUPAC strings (*SoxOct* = CWTTSTHATGCAAAT and *MORE* = ATGMATATKCAT) allowing for one mismatch per 6 bp.

### EMSA

EMSAs were performed essentially as described [27] using 100 nM of 5′Cy5-labeled *SoxOct* or *MORE* dsDNA mixed with 100–500 nM of Oct POUs and 20–100 nM of Sox2 HMG (79-aa protein). Reaction mixtures were incubated for 1 hr on ice in the dark. For single-tube EMSAs, FAM-labeled *MORE* DNA was used. Protein–DNA complexes were separated on native pre-run 1× Tris–glycine native gels at 200 V for 15–30 min. Gel images were taken using Typhoon 9140 PhosphorImager (GE Healthcare) with 500–700 PMT voltage at 50-μm pixel-size resolution with platen focal plane. The relative abundance of each possible microstate was quantified using Image-Quant TL software (Amersham Biosciences) with rubber band option as background correction with a box size of 80 per lane. At least three replicates were performed to calculate omega values. To ensure omega values are estimated under equilibrium conditions, only lanes where all microstates (dimers, monomers, and free DNA) were clearly detectable were used for the calculations. Cooperativity factors (ω) were calculated after quantifying the fraction of bound DNA in EMSAs using the below equations. Equations were derived using Boltzmann weights and principles of statistical mechanisms as detailed in [27,40].

Heterodimer cooperativity:

$$\omega = \frac{K_{d1}}{K_{d21}} = \frac{K_{d2}}{K_{d12}} = \frac{f_0 f_3}{f_1 f_2};$$

where $K_{d1}$ and $K_{d2}$ are the equilibrium binding constants for the proteins 1 and 2 to the composite DNA element alone and $K_{d21}$ and $K_{d12}$ the equilibrium binding constants of binding of proteins 1 and 2 to the DNA element in the presence of the respective other protein. $f$ denotes the bound fraction of the DNA as dimer ($f_3$), monomer 1 ($f_1$), monomer 2 ($f_2$), and the free DNA ($f_0$).

Homodimer cooperativity:

$$\omega = \frac{K_{d1}}{K_{d11}} = \frac{4 f_0 f_2}{f_1^2};$$

where $K_{d1}$ and $K_{d11}$ are equilibrium binding constants for the binding of protein 1 to a DNA element with two of its binding elements as monomer or dimer, respectively. $f$ denotes the fraction of bound DNA for the homodimeric state ($f_2$), the dimeric state ($f_1$), and the free DNA ($f_0$).

Sequences of EMSA oligonucleotides are in Appendix Table S1.

### EMSAs to monitor transcription factor–DNA dissociation

The buffer system, proteins, and DNA elements are identical to the EMSAs performed under equilibrium conditions. The lid of the mini gel chamber (Bio-Rad) was removed to allow for sample loading without powering off the gel. The binding reaction of the POU domains and labeled reporter DNA was prepared in a 220 μl reaction volume and incubated for 1 h for the reaction to reach equilibrium. Subsequently, unlabeled competitor DNA was added from a 100 μM stock and 20 μl aliquots were removed from the reaction mix after different time intervals and loaded onto running gels. 10% native PAGE gels were used for the experiment and run at 100 V for 1.5 h. Gels were imaged using a FLA-7000 image reader (FUJIFILM).

### Structural modeling of DNA-bound homodimers

We built 1,250 homology models for Oct4 and Oct6 homodimers using MODELLER 9.17 (https://salilab.org/modeller/) and the

following templates (referred by PDB ID): (i) 2xsd—crystal structure of Oct6 homodimer bound to *MORE* [42]; (ii) 1e3o—crystal structure of Oct1 homodimer bound to *MORE* [17]; (iii) 3l1p—crystal structure of Oct4 homodimer bound to a palindromic site different than *MORE* [44]; (iv) 1gt0—crystal structure of the Oct1–Sox2–DNA ternary complex [21]. We imposed symmetry restraints on Cα, Cβ, and Cγ atoms to enforce a similar structure of the two monomers. We transferred the DNA as a rigid body (using special restraints) from the 1e3o structure. We used a "slow" optimization protocol followed by a "slow" molecular dynamics-based refinement protocol. The linker region (residues 76–96) was modeled using "loop-model" without imposing any restraints on its structure. The best three models of each homodimer were selected based on a normalized energy score. For the Oct4 models, the scores were −0.8621, −0.8473, and −0.8433, whereas for Oct6, the scores were −0.9331, −0.9229, and −0.9125 (the score ranging from +2 to −2). Then, we extended DNA at each end with 5 base pairs to avoid potential simulation artifacts due to a too short DNA length in the crystal structure and adapted the sequence of DNA in CHIMERA to obtain the final sequence in the models: 5′-CACAGTGC*TCATGCATATGCATGA* GCCTGGGA-3′ (*MORE* motif highlighted).

## MD simulations

Ionizable residues were assigned their standard protonation state at neutral pH. The 5′ and 3′ ends of the DNA and the N-termini of the proteins were methylated, whereas the C-termini of the proteins were acetylated to avoid potential truncation artifacts. Then, the systems were (i) solvated in a truncated octahedral box of SPCE water extending at least 12 Å from any protein/DNA atom, (ii) neutralized with 46 Na$^+$ ions and additional 100 mM KCl (135 ions) using the Merz ions [59] to mimic the experimental ionic strength. We used the Amber-ff14SB [60] and the Amber-parmbsc1 [61] force fields for proteins and DNA, respectively. First, the energy of the systems was minimized in 11 steps as described in Appendix Table S2. Then, the systems were equilibrated for 10.3 ns in 15 steps as described in Appendix Table S3. Finally, each system was simulated in NPT ensemble for 200 ns with a timestep of 2 fs resulting in ensembles of 600 ns (from the three selected models) for each Oct4 and Oct6 homodimers bound to *MORE* DNA. The temperature was maintained at 300 K with Langevin dynamics with a damping coefficient of 0.1/ ps. The pressure was maintained at 1 atm with the Nose Hoover Langevin piston method with the period and decay of 1.2 and 1.0 ps, respectively. The direct calculation of the non-bonded interactions was truncated at 10 Å. Long-range electrostatics were calculated using the particle mesh Ewald algorithm [62]. A correction to the energy and the pressure was also applied to account for the truncated long-range Lenard-Jones interactions [63]. The length of the atomic bonds involving hydrogen atoms was constrained with the SHAKE [64] and the SETTLE [65] algorithms for the macromolecule and water, respectively. All simulations were performed in NAMD 2.11 [66]. Snapshots were selected for analysis every 4 ps.

## Site-directed mutagenesis

Mutations into full-length CDS of the mouse Oct4 and Oct6 proteins (the Consensus CDS Database IDs are 37600.1 and 57296.1, respectively) were made in the pMX cloning vector using the QuikChange

Site-Directed Mutagenesis Kit according to the manufacturer's protocol (Agilent Technologies). Specific oligos used for each modification of DNA sequence are listed in Appendix Table S4. Mutant plasmids were selected after DNA sequencing.

## Cell culture

HEK293T cells, mouse embryonic fibroblasts (MEFs), and OG2 MEFs were cultured in low-glucose (1,000 mg/l) Dulbecco's modified Eagle's medium (DMEM, Sigma-Aldrich), supplemented with 10% fetal bovine serum (FBS, Biochrom), 2 mM L-glutamine and 1× penicillin/streptomycin (Sigma-Aldrich), 1% non-essential amino acids (Sigma-Aldrich), and 0.10 mM β-mercaptoethanol. OG2-MEFs were isolated as described previously [1]. Mouse embryonic stem cells (mESCs) and iPSCs were maintained in high-glucose (4,500 mg/l) DMEM (Sigma-Aldrich), supplemented with 10% fetal bovine serum (Biochrom), 5% serum replacement (Gibco), 2 mM L-glutamine, 1× penicillin/streptomycin (Sigma-Aldrich), 1% non-essential amino acids (Sigma-Aldrich), 1% sodium pyruvate (Sigma-Aldrich), 0.10 mM β-mercaptoethanol (Gibco) with 1,000 units of leukemia inhibitory factor (LIF; prepared in house) on a feeder layer of gamma-irradiated MEFs; experiments under feeder-free conditions were performed using mESC medium and 2,000 units of LIF, as previously described [67].

## Virus production, induction of pluripotent stem cells

Prior to transfection, HEK293T cells were seeded on 100-mm dishes. The following day, the pMX retroviral vectors containing wild-type Oct4, Oct6, Sox2, Klf4, and c-Myc as well as mutated cDNAs for mouse Oct4 or Oct6 were co-transfected with packaging helper plasmid pCL-Eco into $2 \times 10^6$ HEK293T cells using the Fugene6 transfection reagent (Roche). The medium was changed on the next day. 48 hours after the infection, virus-containing supernatants were collected, filtered (Millex-HV 0.45 μm; Millipore), supplemented with 6 μg/ml protamine sulfate (Sigma-Aldrich), and used directly for infection. OG2-MEFs were plated on six-well plates at a density of $2.5 \times 10^4$ cells per well, grown for 24 h, and transduced twice in 24-h intervals with equal amount of each virus-containing supernatant. The medium was changed to mESC medium 1 day after the second infection. ESC medium was changed every second day. Reprogramming experiment was repeated three times, and the GFP colonies were counted 16 days after the second infection under a fluorescent microscope.

## Oct4 complementation assay

The ZHBTc4 embryonic stem cells were infected with WT Oct4, WT Oct6, and Oct6$^{7K,22T,LinkO4,151S}$ using lentiviral supernatants following procedures established by Niwa *et al* [68]. 24 hours after the infection, doxycycline was added to the ESC medium (final concentration: 1 μg/1 ml), in order to suppress the endogenous Oct4, as described in [68]. The self-renewing rescued ES colonies were maintained in culture for further analysis by qRT–PCR.

## qRT–PCR and microarray

RNA samples to be analyzed by qRT–PCR and microarrays were prepared using RNeasy Mini Kit (QIAGEN) with on-column DNA

digestion. Complementary DNA for qRT–PCR was synthesized with the M-MLV Reverse Transcriptase (Affymetrix). Transcript levels were determined using ABI PRISM Sequence Detection System 7900, and gene expression was normalized to the housekeeping gene *Gapdh*. Specific qRT–PCR primers are listed in Appendix Table S5.

For microarray analysis, 300 ng of total RNA per sample was used as input using a linear amplification protocol (Ambion), which involved synthesis of T7-linked double-stranded cDNA and 12 h of *in vitro* transcription incorporating biotin-labeled nucleotides. Purified and labeled cDNA was then hybridized for 18 h onto MouseRef-8 v2 expression BeadChips (Illumina) following the manufacturer's instructions. After washing, chips were stained with streptavidin-Cy3 (GE Healthcare) and scanned using the iScan reader (Illumina). At least two independent iPSC lines were analyzed.

The data discussed in this publication have been deposited in NCBI's Gene Expression Omnibus [69] and are accessible through GEO Series accession number GSE81908 (https://www.ncbi.nlm.nih.gov/geo/query/acc.cgi?acc = GSE81908).

### Genotyping

Genomic DNA was purified using QIAamp DNA Mini Kit (QIAGEN) and amplified using the GoTaq DNA Polymerase (Promega), and specific PCR primers are listed in Appendix Table S5. PCR products were analyzed using 2% agarose gels.

### In vitro differentiation of mouse iPSCs

Embryoid bodies (EBs) were generated from mouse iPSCs via the hanging-drop method, with $1 \times 10^3$ cells in a 30 μl drop of mouse ESC medium without LIF. EBs were gently collected after 5 days and plated on gelatinized 6-well plates at a density of 20 EBs per well. EBs were cultured for 3 weeks and LIF-negative ESC medium was changed every 3 days. After spontaneous differentiation, structures of all three germ layers were observed, and immunostaining was performed.

### Teratoma formation

About $5 \times 10^6$ iPSCs were injected subcutaneously into the flanks of severe combined immunodeficiency (SCID) mice. After 4–5 weeks, mice were sacrificed and the teratoma that had formed was excised, fixed in 4% paraformaldehyde, stained with hematoxylin and eosin, and subjected to histological examination.

### Immunochemistry

The cells were fixed by incubation in 4% (v/v) paraformaldehyde in phosphate-buffered saline (PBS) for 20 min at room temperature (RT) and then rinsed three times with PBS. Cells were permeabilized by incubation in 0.1% Triton X-100 in PBS for 15 min at RT and then washed three times with PBS. Cells were blocked in 5% BSA and 1% goat serum in PBS for 1 h at RT. Primary antibodies—goat polyclonal anti-Sox2 (Y17; Santa Cruz; 1:400), rat monoclonal anti-Nanog (eBioscience; 1:1,000), mouse monoclonal anti-smooth muscle actin (SMA) (Sigma-Aldrich; 1:500), goat polyclonal anti-α-fetoprotein (AFP) (C-19; Santa Cruz; 1:400), mouse monoclonal anti-β-tubulin III (Sigma-Aldrich; 1:800)—were diluted in 1% BSA in PBS and applied at 4°C for overnight. The cells were washed with

PBS three times for 5 min. Alexa Fluor 568, 647, fluorophore-conjugated secondary antibodies (Invitrogen) were diluted 1:1,000 in 1% BSA/PBS. Secondary antibodies were applied for 2 h at RT in the dark and cells were subsequently washed with PBS three times for 5 min. Finally, cells were incubated for 10 min in 300 ng/ml DAPI/PBS and rinsed thrice with PBS for 1 min. Samples were later visualized using a fluorescent microscope.

### Karyotyping of mouse iPSCs

Mouse iPSCs were cultured in a 3-cm dish and the medium was changed 12 h before starting the protocol. To obtain single cells, cells were trypsinized and kept on a gelatinized dish for 40 min to quickly remove attached MEFs. Any remaining big clumps of MEFs were removed by using a 40-μm filter. Cells were transferred to a 15-ml tube and incubated in 2 ml DMEM with 0.5 μg/ml nocodazole for 2 h. Next, cells were centrifuged at $1 \times 10^3$ rpm for 4 min, and the cell pellet was resuspended in 100 μl medium. 1 ml of prewarmed 0.56% KCl was slowly added drop by drop to a total volume of 3 ml. The cell suspension was incubated for 12 min at 37°C in a water bath, and cells were pelleted by centrifugation ($1 \times 10^3$ rpm, 5 min) and resuspended in 100 μl 0.56% KCl. After 1 min, a fresh fixative (methanol/acetic acid at a ratio of 3:1) was slowly added, starting from 5 μl up to 3 ml. The cells were incubated at RT for 30 min, washed three times with 2 ml of fixative, and resuspended in 50–300 μl of the fixative. Large cell clusters were allowed to disintegrate for 1 min and one drop was placed on a dry cover slide and left for at least 30 min to air-dry. Samples were stained with DAPI and examined for metaphase plates.

### Bisulfite sequencing

To determine DNA methylation status at regulatory regions of *Oct4*, *Nanog*, and *Col1a1*, bisulfite conversion was carried out on 2 μg of isolated genomic DNA from cells with the EZ DNA methylation kit (Zymo research) according to the manufacturer's protocol. The bisulfite-converted DNA was amplified by PCR with previously described primers [70,71]. The PCR products were cloned into the pCRII TOPO vector (Invitrogen), and plasmids extracted from individual clones were sequenced by GATC-biotech (http://www.gatc-biotech.com/en/index.html). Sequences were analyzed using the Quantification Tool for Methylation Analysis (QUMA; http://quma.cdb.riken.jp).

### Chimera generation and germline transmission

All (C57BL/6 × C3H) F1 female mice for embryo collection were treated with 7.5 IU pregnant mare's serum gonadotropin (PMSG) and 7.5 IU human chorionic gonadotropin (HCG) in 48 h apart, and then crossed with CD1 male mice. Eight-cell embryos were flushed from female mice at E2.5 and placed in M2 medium [70]. Trypsin-digested iPSCs (8–12 cells and one embryo per aggregate) were transferred into a depression in the microdrop of KSOM. Meanwhile, batches of 30–50 embryos were briefly incubated in acidified Tyrode's solution [70] until dissolution of their zona pellucida. A single 2n embryo was placed in each depression. All aggregates were cultured overnight at 37°C and 5% $CO_2$. After 24 h of culture,

   

the majority of aggregates had formed blastocysts. 10–14 embryos were transferred into one uterine horn of an E2.5 pseudopregnant recipient. 11 days after embryo transfer, E13.5 fetuses were collected and their gonads were checked for the presence of Oct4-GFP. 34 full-term chimera pups were delivered by natural birth or Cesarean section at E19.5. Of all 34 pups, seven pups died at birth and seven female pups were sacrificed. Thus, 20 male pups were set up with foster mothers to check for germline transmission after sexual maturation.

## Data availability

### Primary data

Stepan Jerabek, Calista K. L. Ng, Guangming Wu, Marcos J. Arauzo-Bravo, Kee-Pyo Kim, Daniel Esch, Vikas Malik, Yanpu Chen, Sergiy Velychko, Caitlin M. MacCarthy, Xiaoxiao Yang, Vlad Cojocaru, Hans R. Schöler and Ralf Jauch (2016) Changing POU dimerization preferences converts Oct6 into a pluripotency inducer. Gene Expression Omnibus GSE81908.

### Referenced data

Heinz S, Benner C, Spann N, Bertolino E, Lin YC, Laslo P, Cheng JX, Murre C, Singh H, Glass CK (2010) Simple combinations of lineage-determining transcription factors prime cis-regulatory elements required for macrophage and B-cell identities. Gene Expression Omnibus GSE21512.

Wapinski OL, Vierbuchen T, Qu K, Lee QY, Chanda S, Fuentes DR, Giresi PG, Ng YH, Marro S, Neff NF, Drechsel D, Martynoga B, Castro DS, Webb AE, Sudhof TC, Brunet A, Guillemot F, Chang HY, Wernig M (2013) Hierarchical mechanisms for direct reprogramming of fibroblasts to neurons. Gene Expression Omnibus GSE43916.

Lodato MA, Ng CW, Wamstad JA, Cheng AW, Thai KK, Fraenkel E, Jaenisch R, Boyer LA (2013) SOX2 co-occupies distal enhancer elements with distinct POU factors in ESCs and NPCs to specify cell state. Gene Expression Omnibus GSE35496.

Marson A, Levine SS, Cole MF, Frampton GM, Brambrink T, Johnstone S, Guenther MG, Johnston WK, Wernig M, Newman J, Calabrese JM, Dennis LM, Volkert TL, Gupta S, Love J, Hannett N, Sharp PA, Bartel DP, Jaenisch R, Young RA (2008) Connecting microRNA genes to the core transcriptional regulatory circuitry of embryonic stem cells. Gene Expression Omnibus GSE11724.

Remenyi A, Tomilin A, Pohl E, Lins K, Philippsen A, Reinbold R, Scholer HR, Wilmanns M (2001) Differential dimer activities of the transcription factor Oct-1 by DNA-induced interface swapping. Protein Data Bank 1E3O.

Remenyi A, Lins K, Nissen LJ, Reinbold R, Scholer HR, Wilmanns M (2003) Crystal structure of a POU/HMG/DNA ternary complex suggests differential assembly of Oct4 and Sox2 on two enhancers. Protein Data Bank 1GT0.

Jauch R, Choo SH, Ng CKL, Kolatkar PR (2011) Crystal structure of the dimeric Oct6 (Pou3f1) POU domain bound to palindromic MORE DNA. Protein Data Bank 2XSD.

Esch D, Vahokoski J, Groves MR, Pogenberg V, Cojocaru V, Vom Bruch H, Han D, Drexler HC, Arauzo-Bravo MJ, Ng CK, Jauch R, Wilmanns M, Scholer HR (2013) A unique Oct4 interface is crucial for reprogramming to pluripotency. Protein Data Bank 3L1P.

**Expanded View** for this article is available online.

## Acknowledgements

R.J. is supported by a 2013 MOST China-EU Science and Technology Cooperation Program (Grant No. 2013DFE33080), MOST grant 2016YFA0100700, the National Natural Science Foundation of China (Grant No. 31471238), a 100 talent award of the Chinese Academy of Sciences, and a Science and Technology Planning Project of Guangdong Province, China (2014B030301058, 2016A050503038). V.C. is supported with funding and computer resources by the Max Planck Society and the German Research Foundation (Grant Number CO 975/1-1). S.J. thanks the Cells in Motion (CiM) Graduate School and the International Max Planck Research School—Molecular Biomedicine (IMPRS-MBM) Joint Graduate Program for support. V.M. is supported by a CAS-TWAS President's Fellowship of the University of the Chinese Academy of Sciences (UCAS) and The World Academy of Science (TWAS).

## Author contributions

SJ and CKLN designed and performed the experiments and analyzed the data. GW performed *in vivo* assays. MJA-B analyzed the microarray data. K-PK performed the DNA methylation analysis. DE helped with experimental design, cloning, and reprogramming. VM and RJ performed the ChIP-Seq data analysis. YC performed the kinetic EMSAs. CMM helped with preparation of material for the kinetic EMSAs. SV contributed to the reprogramming data set. XY validated reprogramming experiments and analyzed data. VC performed and interpreted the structural models and simulations and contributed to the design of the study. HRS and RJ obtained funding and designed the study. SJ and RJ wrote the manuscript.

## Conflict of interest

The authors declare that they have no conflict of interest.

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
