## [Review Process File · EMBO Reports]

Manuscript EMBO-2016-42958

Changing POU dimerization preferences converts Oct6 into a pluripotency inducer

Stepan Jerabek, Calista K. L. Ng, Guangming Wu, Marcos J. Arauzo-Bravo, Kee-Pyo Kim, Daniel Esch, Vikas Malik, Yanpu Chen, Sergiy Velychko, Caitlin M. MacCarthy, Xiaoxiao Yang, Vlad Cojocaru, Hans R. Schöler, and Ralf Jauch

Corresponding authors: Hans R. Schöler, Max Planck Institute for Molecular Biomedicine; Ralf Jauch, Guangzhou Institutes for Biomedicine and Health, Chinese Academy of Sciences

Review timeline:

Submission Date:	26 June 2016
Editorial Decision:	15 July 2016
Revision Received:	13 October 2016
Editorial Decision:	25 October 2016
Revision Received:	02 November 2016
Accepted:	08 November 2016

Editor: Achim Breiling

Transaction Report:

1st Editorial Decision

15 July 2016

Thank you for the submission of your research manuscript to EMBO reports. We have now received reports from the three referees that were asked to evaluate your study, which can be found at the end of this email.

As you will see, all referees acknowledge the high interest of the findings and support publication of the study. However, referee #2 has raised several concerns that I ask you to fully address in a complete point-by-point response. Acceptance of your manuscript will depend on a positive outcome of a second round of review. It is EMBO reports policy to allow a single round of revision only and acceptance or rejection of the manuscript will therefore depend on the completeness of your responses included in the next, final version of the manuscript.

Revised manuscripts should be submitted within three months of a request for revision; they will otherwise be treated as new submissions. Please contact us if a 3-months time frame is not sufficient for the revisions so that we can discuss the revisions further.

REFeree REPORTS

Referee #1:

The manuscript by Jerabek et al., reports a detailed study aimed at identifying molecular features defining functional specificity distinguishing Oct4 and Oct6. In this study, the authors first provide a series of experiments showing that Oct4 and POU somatic factors exhibit intrinsic preference for binding to SoxOct or MORE elements respectively. Based on structural models they demonstrate that the predicted Ser151 and the corresponding Met151 are crucial to allow Oct4 and Oct6 binding sequence specificity.

Then the authors identified Oct4 "elements" required for efficient induction of pluripotency. Through the generation of a battery of mutants, the authors identify specific aminoacids or the POU linker domain as relevant for efficient induction of pluripotency. Then since Oct6 cannot induce pluripotency, the authors analyze whether replacement of Oct4 elements into Oct6 are sufficient to convert Oct6 into a pluripotency inducing factor.

This experiment shows that the four elements together enabled Oct6 to induce pluripotency. In the last part of the study the authors demonstrate that Oct6 mutant iPSCs are genuine iPSCs. I consider this as an important study precisely addressing functional features defining binding specificity of POU factors as well as Oct4 -dependent inducing pluripotency ability. In addition the experiments are logic and well designed. In sum I support the publication of this manuscript in EMBO Reports.

Referee #2:

This paper presents an interesting study of the molecular determinants of Oct TFs' pluripotency-inducing capability in terms of their different propensities for the homodimeric or heterodimeric forms. The authors found that a single point mutation is sufficient to shift the equilibrium position between the two forms, and when combined with two more mutations could turn Oct6 into a pluripotency inducer. The biological significance of the discovery, as the authors rightly pointed out, lies in the fact that "subtle modifications at the molecular interfaces... can profoundly swap their lineage specifying activities." Overall, the data presented are convincing, and the discoveries of significance to the field of stem cell biology, reprogramming and TF biology. It would be important if the authors could address these points:

1) The study falls a little short on deeper mechanistic investigation behind the remarkable equilibrium-altering capability of this single-point mutation. Given that the mutation, which is a major result in this study, was discovered based on careful analysis of structural information, it would be more convincing if the authors could again show or refer to structural data to validate their finding, i.e. show that the S151M or M151S mutation on Oct4 or Oct6 indeed leads to significant structural changes at the contact site with POUHD, thereby elucidating the mechanistic basis for the changes in affinity of the TFs for the homo- or heterodimeric form. The same could be said for the 7K/22T double mutation.

2) While the authors have demonstrated that changing the dimerization preferences of Oct TFs can alter their lineage-specifying capability, one could not help but wonder if dimerization represents the whole picture. Could the rate of dimer formation or the time during which they stay in the homo- or heterodimeric form also contribute to their pluripotency-inducing capability? Such possibilities need to be examined or discussed.

3) Page 5: Please explain what is special about 'HOMER', and why is it capable of discovering previously hidden features from publically available ChIP-Seq datasets?

4) Page 6: The cooperativities of the 6 POU TFs are quantified "using previously derived equations". More details are needed here to justify the analysis. Also, do the "cooperativity" values shown in Fig 1D and E refer to the Hill coefficient? If so, the values of 100-250 as reported in Fig

1D seems way too high... Moreover, the values in Fig 1E are more than 1 order of magnitude smaller than those in 1D. For the same TF to exhibit such drastic changes in cooperativity seems highly unusual, leading to questions on the validity of such quantitative analysis.

5) Fig 3B and C: The validity of the quantification here rests solely on the accurate counting of MEF colonies, but judging from the images shown in Fig 3C, this does not seem as straightforward as it sounds, since how one defines a "colony" in such images could at times be rather arbitrary. For example, how many colonies are there in the Oct4WT image? And for Oct4151M, is there 1 or 3 colonies? The authors need to provide details to demonstrate that such counting is done in an accurate and unbiased fashion.

6) Fig 4A: The 7K/22T mutation introduced here seems very abrupt, since it was not tested in the previous Figure and has not been mentioned before. How did this mutation come about? And why are the amino acids K and T chosen for these two positions?

7) There are multiple issues with phrasing and wording, grammar, labeling, spelling, unclear expression, etc. For example:

The first sentence of Introduction begins with the word "already", which is grammatically awkward

In Fig 2A legend: "...Oct-Oct (left) and Oct6-Oct6 (right)". I assume it should read "(top) and (bottom)"? Each subpanel should be clearly labeled in the Figure to make it easier for the reader. The same applies to Figs 1B and 3A.

Fig 4E: In label "DNA (Dapi)", DAPI should be capitalized throughout

The beginning of the Abstract gives no background, and thus seems very abrupt. The reference list uses a different font from the main text.

Referee #3:

This is a phenomenal manuscript dissecting with unprecedented detail and elegance residues dictating pro-pluripotency dictating ability of Oct4 vs Oct6 transcription factors. Not only the authors were able to hamper Oct4 pro-pluripotency ability with a single mutation, but they "Reprogrammed" Oct6 to become a pro-pluripotency factor and generate high quality iPSCs. The rational leading the authors to make this discovery pinpoint OctSox dimer ability as key determinant for pro-pluripotency function of Pou family members of TFs. The manuscript is well written, the figures are elegant and fully support the conclusions made. Methods are detailed, and references are adequate and unbiased. I have no comments on how to further improve this outstanding work.

1st Revision - authors' response

13 October 2016

Referee #1:

The manuscript by Jerabek et al., reports a detailed study aimed at identifying molecular features defining functional specificity distinguishing Oct4 and Oct6. In this study, the authors first provide a series of experiments showing that Oct4 and POU somatic factors exhibit intrinsic preference for binding to SoxOct or MORE elements respectively. Based on structural models they demonstrate that the predicted Ser151 and the corresponding Met151 are crucial to allow Oct4 and Oct6 binding sequence specificity.

Then the authors identified Oct4 "elements" required for efficient induction of pluripotency. Through the generation of a battery of mutants, the authors identify specific aminoacids or the POU linker domain as relevant for efficient induction of pluripotency. Then since Oct6 cannot induce pluripotency, the authors analyze whether replacement of Oct4 elements into Oct6 are sufficient to convert Oct6 into a pluripotency inducing factor.

This experiment shows that the four elements together enabled Oct6 to induce pluripotency. In the last part of the study the authors demonstrate that Oct6 mutant iPSCs are genuine iPSCs.

I consider this as an important study precisely addressing functional features defining binding specificity of POU factors as well as Oct4 -dependent inducing pluripotency ability. In addition the experiments are logic and well designed. In sum I support the publication of this manuscript in EMBO Reports.

Response: We thank the reviewer for his/her positive assessment of our manuscript.

Referee #2:

This paper presents an interesting study of the molecular determinants of Oct TFs' pluripotency-inducing capability in terms of their different propensities for the homodimeric or heterodimeric forms. The authors found that a single point mutation is sufficient to shift the equilibrium position between the two forms, and when combined with two more mutations could turn Oct6 into a pluripotency inducer. The biological significance of the discovery, as the authors rightly pointed out, lies in the fact that "subtle modifications at the molecular interfaces... can profoundly swap their lineage specifying activities." Overall, the data presented are convincing, and the discoveries of significance to the field of stem cell biology, reprogramming and TF biology. It would be important if the authors could address these points:

1) The study falls a little short on deeper mechanistic investigation behind the remarkable equilibrium-altering capability of this single-point mutation. Given that the mutation, which is a major result in this study, was discovered based on careful analysis of structural information, it would be more convincing if the authors could again show or refer to structural data to validate their finding, i.e. show that the S151M or M151S mutation on Oct4 or Oct6 indeed leads to significant structural changes at the contact site with POUHD, thereby elucidating the mechanistic basis for the changes in affinity of the TFs for the homo- or heterodimeric form. The same could be said for the 7K/22T double mutation.

Response: We thank the reviewer for this comment. To further understand the mechanism by which M151 in Oct6 and S151 in Oct4 affect the homodimerization, we performed classical molecular dynamics simulations for the Oct4 and Oct6, homodimers on MORE DNA. For each homodimer, we performed 3 independent simulations, each of which was 200 ns long, starting with three different initial models obtained from homology modeling (in total 600 ns per system). The models differed in the linker conformation which is unknown for the MORE-bound configuration. We observed significant differences between Oct4 and Oct6 both in terms of global dynamics as well as regarding the detailed structural environment of the residue 151. These differences provide further explanations on why the homodimer interface in Oct4 is less optimal than in Oct6. Because the residues surrounding residue 151 are very well conserved between Oct4 and Oct6, simulations of the mutants are unlikely to bring further insights. Therefore, and also considering that simulations are time consuming and computationally expensive, we limited ourselves to simulations of the wild types. The results are now shown in new panels in main Fig. 1G and Fig. 2B, 2C.

We agree with the reviewer that it would be fantastic to also explain the mechanism by which residues K7 and T22 function. However, investigating the structures and simulations available (from this study and from our previous studies by Merino et al. 2014 [1] and Merino et al. 2015 [2]) we cannot draw clear conclusions on how these residues contribute to the function of Oct4. T22 is close to the Oct-Sox interface but does not have a major contribution to the interaction, whereas K7 is at the beginning of helix 1 of the POU_S domain without being involved in any known interaction of Oct4. Therefore, further studies will be needed to clarify the role of these residues.

We describe the data in a new paragraph in the Results section pages 6 and 7 "2. The Oct4 homodimer is unstable and structurally flexible", with relevant Fig. 1G, Fig. 2B and 2C. Materials and Methods sections named "Molecular dynamics simulations", "Building structural models of DNA-bound Oct4 and Oct6 homodimers", "System preparation for MD simulations." and "MD simulations" relate to the new chapter, as well as Appendix Table 2 and 3.

2) While the authors have demonstrated that changing the dimerization preferences of Oct TFs can alter their lineage-specifying capability, one could not help but wonder if dimerization represents the whole picture. Could the rate of dimer formation or the time during which they stay in the homo- or heterodimeric form also contribute to their pluripotency-inducing capability? Such possibilities need to be examined or discussed.

Response: We agree that not only equilibrium binding determines the regulatory outcome of a TF-DNA interaction. Rather, the binding kinetics of TF-DNA interactions can be of critical importance. As suggested by the reviewer we examined the “time during which they [the TFs] stay in homo- or heterodimeric form” using newly designed time-resolved competition EMSAs. Indeed, these experiments reveal profound differences in the dissociation kinetics between Oct4 and Oct6. We show the new set of experiments in Fig. EV2 and describe the kinetics EMSAs along with the MD experiments in a new paragraph on pages 6 and 7 entitled: “2. The Oct4 homodimer is unstable and structurally flexible”.

3) Page 5: Please explain what is special about 'HOMER', and why is it capable of discovering previously hidden features from publically available ChIP-Seq datasets?

Response: We thank the reviewer for pointing out that we could mislead readers by overemphasizing that only the HOMER software could detect the MORE motif. In the revised version, in addition to the position weight matrix scanning approach, we performed an additional word search using IUPAC strings which gave the same results (enrichment of the MORE motif in POU sites in somatic cells and enrichment of the SoxOct motif in ESCs). We have added this alternative way to detect enriched sequences to Fig. EV1A. In sum, we believe previous studies have not reported the MORE for three main and related reasons:

- (i) Matching de novo motif finding results with known motif databases do not provide unambiguous results with default settings.*
- (ii) Investigators have not specifically looked for the MORE and did not notice cryptic versions of the MORE.*
- (iii) Without instructing de novo motif-finding software to search for longer composite motifs the MORE is not clearly detectable.*

The MORE sequence can escape detection without using longer motif length option (-len 12,14,16 while the HOMER default motif length is -len 8,10,12) during de novo motif finding. For example, a shortened version of MORE is by default reported as Pit1 motif using the HOMER ‘known motif’ function which compares motifs discovered de novo with known motifs deposited in databases such as JASPAR (Fig. 1A; MEFs 48hrs in Brn2 shows shorter version of MORE while in mNPCs shows the full version of MORE). Hence, motif discovery software will not automatically refer investigators to the MORE even if MORE-like sequences are discovered de novo. In fact, some of the publications reporting data we have re-analyzed do in fact report degenerate and shortened versions of the MORE but authors do not refer to it as such. For example, in Wapinski et al. [4], the authors do show a different form of MORE motif in Fig. 4A as a Brn2 motif in NPCs but due to its ambiguity it is referred to as “POU-like motif” in the main text. Further, in Lodato et al. [5] the MORE motif was shown alongside other POU motifs in supplementary table 7 (sheet no.2, row 22nd) and only octamer motif was represented in Fig. 3B. Those are not mistakes made by the authors of these studies but caused by the subtleties of the differences in the binding elements. The first 8bp of the 12bp MORE has a strong resemblance to the canonical octamer leading which may lead to its classification as simple octamer rather than a MORE (octamer: ATGCAAAT; MORE: ATGCATATGCAT). If the sequences retrieved by de novo motif finding are too short, the difference between MORE and the canonical octamer will be barely detectable.

Therefore, we obtain the best results by instructing HOMER to search for motifs of a certain length (options -len 12,14,16). Moreover, knowledge of the older biochemical and crystallographic such as Tomilin et al. [6] is necessary to appreciate the MORE sequence. Whilst the difference between the octamer motif and the first 8bp of the MORE appears subtle, it leads to a profound change in the configuration of the protein-DNA complex (monomeric POU versus a homodimeric POU, Fig. 1B, 1C). Lastly, the presence of the MORE versus SoxOct needs to be validated by careful motif

scanning analysis or searches with dedicated IUPAC strings (Fig. EV1A). We now modify the text of the first paragraph of the Results section (page 5. "To investigate ...").

4) Page 6: The cooperativities of the 6 POU TFs are quantified "using previously derived equations". More details are needed here to justify the analysis. Also, do the "cooperativity" values shown in Fig 1D and E refer to the Hill coefficient? If so, the values of 100-250 as reported in Fig 1D seems way too high... Moreover, the values in Fig 1E are more than 1 order of magnitude smaller than those in 1D. For the same TF to exhibit such drastic changes in cooperativity seems highly unusual, leading to questions on the validity of such quantitative analysis.

Response: We include a more detailed description how we to quantitate the cooperativity constant ω and include the equation used in an expanded methods section page 17. We do not use the hill coefficient but equilibrium ratios of equilibrium binding constants, which can be inferred directly from the relative abundance of the possible microstates (free DNA, monomerically bound DNA and dimerically bound DNA). These states can be directly measured by densitometric evaluation of bands visible in EMSA gels. The equations were derived previously in collaboration with the mathematician Shyam Prabhakar (Genome Institute of Singapore) using principles of statistical mechanics. The formalism for heterodimeric binding was published by Ng et al. [7] and for homodimeric binding by Baburajendran et al. [8]. Gary Stormo (Washington University) adopted these equations for high throughput approaches to measure TF cooperativity using deep sequencing. Here, microstates in EMSA gels are quantified by counting sequencing reads rather than by measuring band intensities (Stormo et al. [9]). Likewise, a group using microfluidics devices to study the dimerization of nuclear receptors has adopted our equations (Isakova et al. [10]). Further, an increase of the binding constant by several orders of magnitude is well within the range of what others and we have observed previously for other TF dimers. In fact, a recent study on nuclear receptors reported w values significantly higher than ours (Isakova et al. [10]). The homodimer cooperativity for Pit1, Oct1, Oct6 and Brn2 is indeed one order of magnitude higher (Fig. 1D) than their respective heterodimer cooperativity (Fig. 1E). This observation finds further support in our kinetic EMSA assays (new Fig. EV2) demonstrating that the homodimeric complexes are substantially more stable than the heterodimeric complexes.

5) Fig 3B and C: The validity of the quantification here rests solely on the accurate counting of MEF colonies, but judging from the images shown in Fig 3C, this does not seem as straightforward as it sounds, since how one defines a "colony" in such images could at times be rather arbitrary. For example, how many colonies are there in the Oct4WT image? And for Oct4151M, is there 1 or 3 colonies? The authors need to provide details to demonstrate that such counting is done in an accurate and unbiased fashion.

Response: In order to show typical GFP-positive colonies generated by our mutants, in Fig. 3D we used the 10x magnification on the fluorescence microscope. However, when we count colonies, we normally rely on a lower magnification to 2 or 2.5x magnification, which allows us to distinguish colonies more clearly from 'longer distance'. Therefore, we now add a new illustrative Appendix Fig. S2B taken with 2.5x lenses that we routinely use for iPSC colony counting. In this setup, one can distinguish a colony by: i) clear spatial separation of the GFP signal and/or ii) morphology of the colony (occasionally, a physical shape of colony in the bright field is helpful if two colonies are too close to each other). Moreover, in the new figure, we indicated counted colonies with dashed white circles. We still like to retain the previous higher magnification images (10x lense) in the main text in Fig. 3D, as it better illustrates the morphology of the colonies. Furthermore, we included an additional replicate to the revised Fig. 3C which now shows the mean +/- standard deviation of three biological replicates and we also performed ANOVA to assess the significance. Last, we included the quantification of all viral titers used in our study, as determined by qRT-PCR. As comparison of reprogramming efficiencies between different mutants may be influenced by relative amount of their viruses in the supernatants, the new Appendix Fig. S2A and Fig. EV4A shows that we used comparable amount of viruses among screened Oct4 and Oct6 mutants.

6) Fig 4A: The 7K/22T mutation introduced here seems very abrupt, since it was not tested in the previous Figure and has not been mentioned before. How did this mutation come about? And why are the amino acids K and T chosen for these two positions?

Response: We wish to point out that this mutant had been characterized in the previous Fig. 3C in the context of Oct4 as well. Nishimoto et al. [11] had identified this mutation as being critical for the maintenance of pluripotency using the Oct4 complementation assays. In the new version of our manuscript, we also perform and present complementation assay (Fig. EV5G, EV5H) as a support that engineered Oct6 can maintain pluripotency of ES cells. Furthermore, we cite Nishimoto's work on several occasions. In the previous paragraph describing mutations introduced in Oct4, we had mentioned that (page 9): "...we chose a double mutant in the first alpha helix of the Oct4 POU₅ subdomain (Oct4^{7D,22K}) previously shown to be required for maintaining pluripotency [9]." We now rephrase and add a further citation to the Nishimoto's study when we describe the engineering of Oct6 (page 10): "However, when 151S was combined with the 7K, 22T double mutant identified by Nishimoto et al [9] to be critical for pluripotency maintenance, we consistently obtained iPSC colonies (Fig. 4A, Fig. EV4A, EV4B)."

7) There are multiple issues with phrasing and wording, grammar, labeling, spelling, unclear expression, etc.

Response: We carefully revised the whole text and made several minor typographical and grammar corrections.

The first sentence of Introduction begins with the word "already," which is grammatically awkward.

Response: We improved the first sentence in the beginning of our Introduction part and now write (page 3): "In 2006, somatic cells were shown to be reprogrammable to pluripotent stem cells by the overexpression of just four transcription factors (TFs)—Oct4, Sox2, Klf4, and c-Myc (OSKM) [10]."

In Fig 2A legend: "...Oct-Oct (left) and Oct6-Oct6 (right)". I assume it should read "(top) and (bottom)"? Each subpanel should be clearly labeled in the Figure to make it easier for the reader. The same applies to Figs 1B and 3A.

Response: Now, we present a new Fig. 2 containing data from molecular simulations (please see our response to the second point of Reviewer #2). We carefully labeled all sub-figures, so the orientation is easy and without any confusion.

Fig 4E: In label "DNA (Dapi)," DAPI should be capitalized throughout

Response: We corrected the label and put the abbreviation of 4',6-diamidino-2-phenylindole -- DAPI -- in capital letters throughout the new version of our manuscript.

The beginning of the Abstract gives no background, and thus seems very abrupt.

Response: We revised our Abstract which now starts with two introductory sentences and believe its readability has improved

The reference list uses a different font from the main text.

Response: We changed the style of text and made the font unified across the re-submitted manuscript.

Referee #3:

This is a phenomenal manuscript dissecting with unprecedented detail and elegance residues dictating pro-pluripotency dictating ability of Oct4 vs Oct6 transcription factors. Not only the authors were able to hamper Oct4 pro-pluripotency ability with a single mutation, but they "Reprogrammed" Oct6 to become a pro-pluripotency factor and generate high quality iPSCs. The rational leading the authors to make this discovery pinpoint OctSox dimer ability as key determinant for pro-pluripotency function of Pou family members of TFs. The manuscript is well written, the figures are elegant and fully support the conclusions made. Methods are detailed, and references are adequate and unbiased. I have no comments on how to further improve this outstanding work.

Response: We are grateful for the encouraging comments of the reviewer.

REFERENCES

1. Merino F, Ng CK, Veerapandian V, Scholer HR, Jauch R, Cojocaru V (2014) Structural basis for the SOX-dependent genomic redistribution of OCT4 in stem cell differentiation. *Structure* 22: 1274-86
2. Merino F, Bouvier B, Cojocaru V (2015) Cooperative DNA Recognition Modulated by an Interplay between Protein-Protein Interactions and DNA-Mediated Allostery. *PLoS Comput Biol* 11: e1004287
3. Lickwar CR, Mueller F, Hanlon SE, McNally JG, Lieb JD (2012) Genome-wide protein-DNA binding dynamics suggest a molecular clutch for transcription factor function. *Nature* 484: 251-5
4. Wapinski OL, Vierbuchen T, Qu K, Lee QY, Chanda S, Fuentes DR, Giresi PG, Ng YH, Marro S, Neff NF, *et al.* (2013) Hierarchical mechanisms for direct reprogramming of fibroblasts to neurons. *Cell* 155: 621-35
5. Lodato MA, Ng CW, Wamstad JA, Cheng AW, Thai KK, Fraenkel E, Jaenisch R, Boyer LA (2013) SOX2 co-occupies distal enhancer elements with distinct POU factors in ESCs and NPCs to specify cell state. *PLoS Genet* 9: e1003288
6. Tomilin A, Remenyi A, Lins K, Bak H, Leidel S, Vriend G, Wilmanns M, Scholer HR (2000) Synergism with the coactivator OBF-1 (OCA-B, BOB-1) is mediated by a specific POU dimer configuration. *Cell* 103: 853-64
7. Ng CK, Li NX, Chee S, Prabhakar S, Kolatkar PR, Jauch R (2012) Deciphering the Sox-Oct partner code by quantitative cooperativity measurements. *Nucleic Acids Res* 40: 4933-41
8. BabuRajendran N, Palasingam P, Narasimhan K, Sun W, Prabhakar S, Jauch R, Kolatkar PR (2010) Structure of Smad1 MH1/DNA complex reveals distinctive rearrangements of BMP and TGF-beta effectors. *Nucleic Acids Res* 38: 3477-88
9. Stormo GD, Zuo Z, Chang YK (2015) Spec-seq: determining protein-DNA-binding specificity by sequencing. *Brief Funct Genomics* 14: 30-8
10. Isakova A, Berset Y, Hatzimanikatis V, Deplancke B (2016) Quantification of Cooperativity in Heterodimer-DNA Binding Improves the Accuracy of Binding Specificity Models. *J Biol Chem* 291: 10293-306
11. Nishimoto M, Miyagi S, Yamagishi T, Sakaguchi T, Niwa H, Muramatsu M, Okuda A (2005) Oct-3/4 maintains the proliferative embryonic stem cell state via specific binding to a variant octamer sequence in the regulatory region of the UTF1 locus. *Mol Cell Biol* 25: 5084-94

2nd Editorial Decision

25 October 2016

Thank you for the submission of your revised manuscript to our editorial offices. We have now received the report from the referee that was asked to re-evaluate your study (you will find enclosed below). As you will see, s/he supports now the publication of your manuscript in EMBO reports. Before we can proceed with formal acceptance, I have a few editorial requests that you need to address in a final revised version of the manuscript.

Please provide for the final version high resolution versions of all main and EV figures in TIFF or EPS format. The current pdf files show compression artifacts and are also rather small.

Further, all materials and methods should be included in the main manuscript file. Therefore please add those from the Appendix to the main manuscript. Maybe this part could also be shortened significantly. Finally, please correct the typo in the TOC of the Appendix (it should be "table of contents").

REFEREE REPORTS

Referee #2:

The authors made appropriate attempts to address each of the points raised. The time-resolved EMSA served their purpose in addressing all remaining concerns. I fully support publication of this study.

2nd Revision - authors' response

02 November 2016

Authors made the requested changes and submitted updated files as needed.

3rd Editorial Decision

08 November 2016

I am very pleased to accept your manuscript for publication in the next available issue of EMBO reports. Thank you for your contribution to our journal.

Corresponding Author Name: Ralf Jauch
Journal Submitted to: EMBO Reports
Manuscript Number: EMBOR-2016-42958-T